# Integrated Solution-Base Isolation and Repositioning-for the Seismic Rehabilitation of a Preserved Strategic Building

**Marco Vailati** [1,*] , **Giorgio Monti** [2] **and Vincenzo Bianco** [2]

1 Department of Civil, Construction-Architectural and Environmental Engineering, University of L'Aquila, Piazzale Ernesto Pontieri, Monteluco, Poggio di Roio, 67100 L'Aquila, Italy
2 Department of Structural Engineering and Geotechnics, Sapienza University of Rome, Via A. Gramsci 53, 00197 Roma, Italy; giorgio.monti@uniroma1.it (G.M.); vincenzo.bianco@uniroma1.it (V.B.)
* Correspondence: marco.vailati@univaq.it

**Abstract:** This paper deals with the design of the seismic rehabilitation of a case-study building located in Florence, Italy. The particular reinforced concrete building hosts an important operational center of the main company that manages the Italian highway network. It is composed of the juxtaposition of three reinforced concrete edifices standing out from a common basement. The design of the interventions for the seismic rehabilitation of this case study posed different challenges, some even in contrast with each other. The main design challenge was to reach the seismic retrofitting, due to the strategic role of the activities hosted herein, safeguarding as much as possible the peculiarity of the architectural elements. Moreover, the design was made harder by the presence of existing thermal joints between adjacent edifices which were inadequate to prevent the latter from pounding upon each other during an earthquake. This outcome yielded the need to intervene by enlarging the gap between the adjacent buildings. This latter intervention was in stark contrast with the explicit request of the client to bring the least possible disturbance to the strategic activities carried out within it; in fact, the joints are crossed by optical fibers and other technological systems which can be damaged easily. The need to fulfill all these design constraints brought the development of an original design strategy based on the employment of base-isolation in a rather unusual configuration. The details of the design procedure, along with the innovative aspects and the designed devices, are presented. With the objective to refine the adopted strategy in view of its possible repeatability by colleague engineers, the paper also presents a fair discussion of every aspect with regards to both the design and the realization phases. Possible ideas for new research and developments are also highlighted.

**Keywords:** base-isolation; lead rubber bearings; strategic building; pounding; strain-induced crystallization; relaxation

## 1. Introduction

The availability of innovative techniques in the structural engineering field nowadays gives designers an opportunity to conceive and realize unprecedented solutions to complex structural problems. This is exactly the case dealt with in this paper, where a seismic rehabilitating intervention was designed for a building located in Florence, Italy. The complexity of the problem resulted from the two-fold role of the building: on one hand, it hosts one of the headquarters of the main Italian Highway Administration Company, which manages a large portion of the Italian highway network, while on the other hand, the building is listed among those protected by the Florentine Cultural Heritage Superintendence.

Among innovative techniques, the implementation of base-isolation systems for the reduction of seismic action on buildings has become a common alternative to conventional strengthening measures [1]. It has been estimated that altogether a total of approximately 16,000 structures have already been protected in different parts of the world by seismic isolation, energy dissipation, and other anti-seismic systems [2]. Most of them are located in Japan, although they are more or less numerous in over 30 countries. A base-isolation

system consists of an isolation layer placed in between the building and the ground, which decouples the building and the soil motion in case of dynamic action. The effectiveness of the base-isolation technique has been extensively investigated in the past for a wide range of structural periods, soil characteristics, and ground motion characteristics. Based on the results of these studies, it is well-acknowledged that seismic isolation is most effective for short-period structures located on firm soil profiles and subjected to far-field ground motions [3–5]. These studies have illustrated that subject to ordinary far-field (non-pulse) ground motions through increasing the structural period, base isolation can effectively reduce the inter-story drift, floor acceleration, and seismic-induced forces in short- to mid-rise structures located on firm-soil profiles [3–10]. For flexible (long-period) structures, structures located on soft-soil profiles, and those subjected to near-filed ground motions, the effectiveness of base isolation depends on many parameters such as the relationship between the periods of the non-isolated structure, the isolated counterpart, the dominant period of the ground motion, and the damping provided by the isolation system. In these cases, seismic isolation could still be an effective approach for seismic protection, but design and application of this technique needs particular care [4,11–16]. Several techniques have been developed that can be used to insert base isolation under existing buildings with some additional costs which might be justified, especially in the case of historic buildings with extremely high or even inestimable value (e.g., [17–22]). Generally, these retrofitting interventions, most of which are realized abroad, end up being very invasive due to the necessity to create a rigid plane both at the intrados and at the extrados of the so-called isolation plane. In fact, in order to fulfill such a need, in most of those cases a thick reinforced concrete slab was realized at the extrados of the isolation layer.

Other techniques go in the direction of controlling the progressive collapse of structural elements via cables located in suitable locations in the structure (e.g., [23]) or use structural control to protect buildings against earthquake excitation (e.g., [24]).

Generally, two types of bearings can currently be found on the market, which are: (1) elastomeric bearings and (2) the so-called friction pendula (e.g., [25]). In particular, in the present work the former technology will be taken into consideration for the reasons explained below.

Possible pounding (e.g., [26,27]) between adjacent edifices not properly separated from each other is another aspect that often has to be faced when dealing with the seismic rehabilitation of existing buildings. It is the case of inadequate width joint among edifices, causing mutual hammering due to out-of-phase oscillation and/or to different fundamental periods of vibration to the opposed buildings.

Due to the consistent budget needs and the technical issues connected to the enlargement of the joints, generally the least expensive solution consists of joining the involved edifices (via mechanical connection that excludes any relative displacement) and studying the behavior under the horizontal force as a single body.

On the contrary, the enlargement of the gap is a very expensive and cumbersome solution due to the necessity to intervene not only on the structure but also on the infills and other non-structural elements. In this latter case it can be useful to employ recent technology which realizes preferential sliding surfaces in the infilled frame—a sort of isolation—whose movement creates a remarkable damping [28].

When designing seismic strengthening projects on buildings of some architectural interest, the main problem is that of reconciling the apparently contradicting requirements of seismically protecting the structures while at the same time safeguarding the figural value of the building. In most cases, this dilemma has been solved so far by resorting to the seismic amelioration solution. In this way, the intervention measures are both non-invasive and respectful of the architectural value, even if a lower seismic protection is eventually achieved. In this case, the so-called "minimum intervention criterion" prevails over the need to pursue a safety level comparable to that of new buildings. However, the question becomes more complex when it is necessary to work on such buildings which also host functions of high strategic importance in addition to boasting a considerable historical

and cultural value. For these buildings, one cannot derogate from the need to pursue full seismic safety.

The above-mentioned seismic amelioration solution is not feasible in these cases. The problem therefore arises of developing a solution able to achieve complete seismic protection of the building without altering, or even slightly modifying, its appearance.

A problem of this kind was presented to the authors when they faced the study of a building that hosts functions of great strategic importance, however, any traditional strengthening intervention on the structural elements would have led to the debasement of the architectural figure.

The paper is subdivided into three sections. The first section describes the building and its main weaknesses. The second part describes in detail the design strategy, providing details of both the adopted solution and designed devices. The third section is devoted to analyzing the design results, highlighting the peculiarity of some technical solutions for possible repeatability by other experts who may work with similar constraints.

## 2. The Case Study

### 2.1. Ante-Operam

The case study building, the subject of the present study, is shown in Figure 1. The plan presents a very squashed Y shape in which the angle between the web and the flange of the Y is approximately 97.2°, so that the obtuse angle between the two flanges is approximately 165.6°. The web is oriented along the north-east-south-west axis and the flanges along the north-west-south-east one. As previously mentioned, the building is composed by the juxtaposition of three Reinforced Concrete (RC) frame edifices standing out from the same basement, namely: (1) the central one, which corresponds to the web of the Y, (2) the eastern, and (3) the western flange (wing). The thermal joint between each of the flanges and the web is large approximately 40 mm.

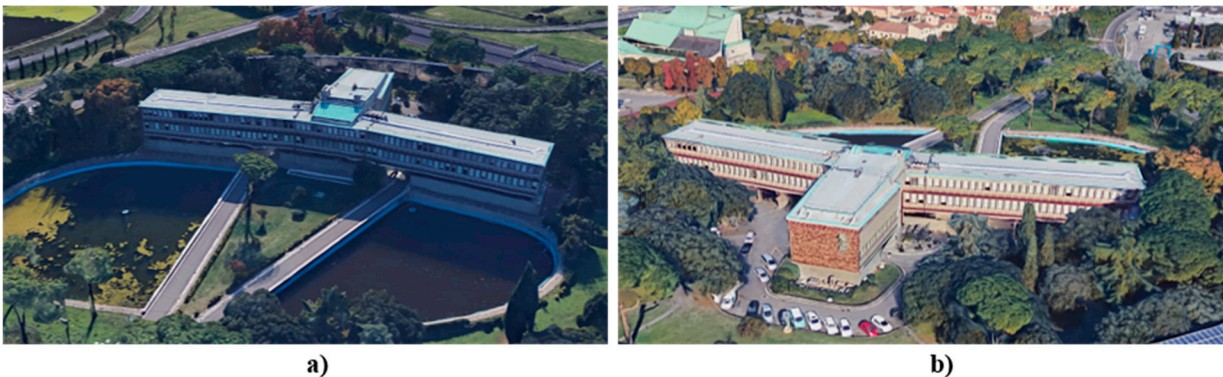

a)      b)

**Figure 1.** Views of the case-study building: (**a**) northern, and (**b**) southern.

The structure is made of RC frames obtained by connecting columns prefabricated on site. A cross section of the building is shown in Figure 2. The building is characterized by a very stiff RC basement, ground floor, and two floors aboveground. The structure of the basement presents very thick perimetral reinforced concrete walls and large central columns, connected by deep beams placed right under the ground floor columns. The ground floor of each of the wings contains 24 RC columns, which have a mushroom shape with a hollow circular shape to accommodate the sewage ducts, as shown in Figure 3. It must be said that the technicians of the Superintendence for the Architectural Heritage of Florence soon underlined the importance of preserving the shape of these columns. However, it can be noted that this is a typical case in which the most distinctive architectural element of the building, namely these ground floor mushroom-shaped columns, are the elements determining the seismic vulnerability of the building. In fact, the presence of the internal hole also produces a considerable reduction of the cross section, which results in a

circular crown whose shear strength is low, especially in the lowermost point of the column. However, it must be remarked that the case study building was designed in the late 1950s. That period was characterized, like anywhere else in the world, by the presence of technical regulations and theoretical notions lacking enough awareness of anti-seismic design.

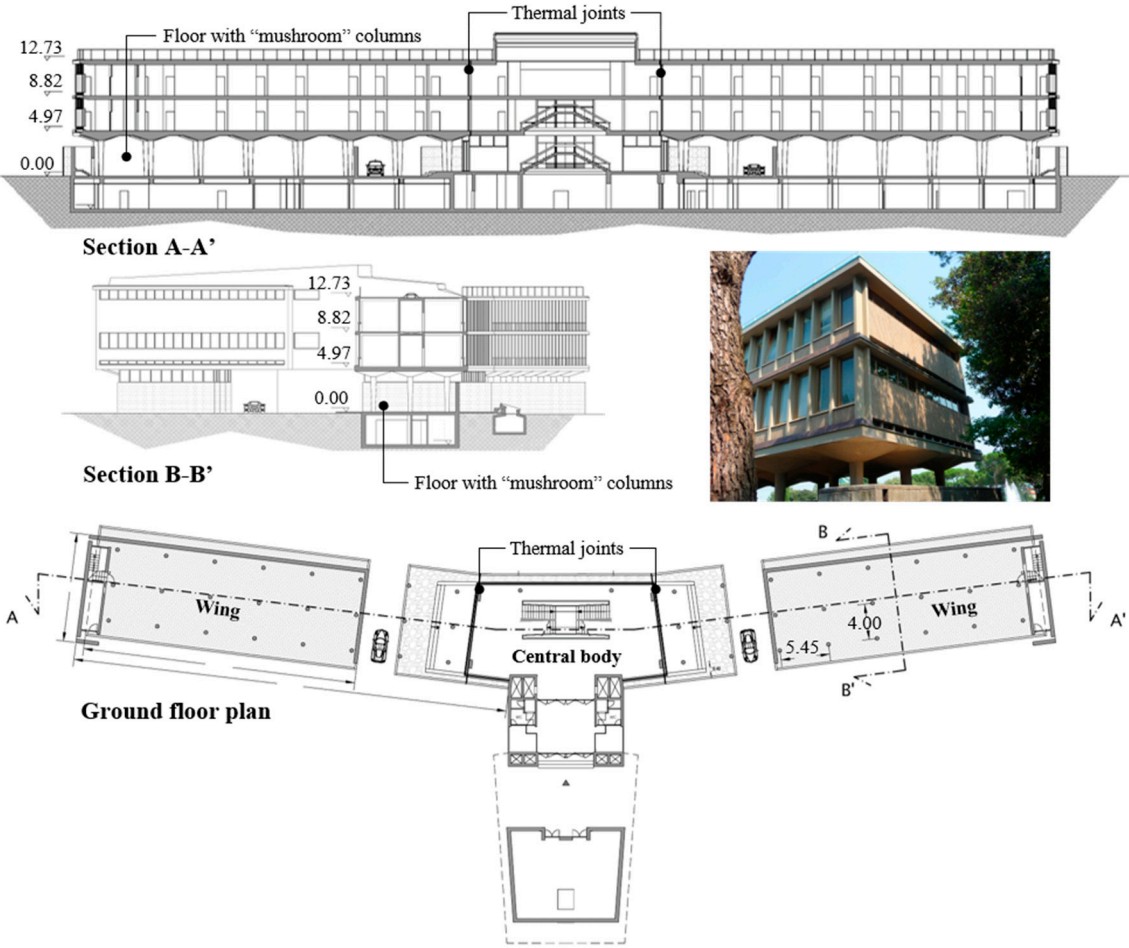

**Figure 2.** Plan, sections, and angular view of the building (all dimensions in m).

It is worth stressing that the ground floor columns have a particular architectural value since they represent a typical symbol of the prefabrication of RC structures, which spread in Italy during the flourishing period of economic growth that followed the second world war (Figure 3). The central building was separated from each of the lateral adjacent edifices by means of a technical joint, probably designed to accommodate essentially thermal expansions, of approximately 40 mm.

### 2.2. Structural Weaknesses

As mentioned in Section 2 the preliminarily evaluations suggest that the ground floor columns might constitute a weak point, due to the contemporary presence of a hollow cross section and a progressive reduction of the cross section along their longitudinal development. The transversal reinforcement was composed of a very thin steel wire, 5 millimeters in. diameter, disposed according to a spiral shape along the height of the column (Figure 4). Moreover, during the in-situ inspections it was possible to realize that these columns also presented a partially spalled concrete cover, with exposed and corroded longitudinal and transversal reinforcement. The presence of corrosion contributed to worsening the already critical shear strength of these columns. The seismic safety evaluation of the building in the ante-operam conditions can be synthetized by the results of the shear check at the base

of the ground floor columns. That being stated, by comparing shear capacity and demand it was possible to verify that the ground floor columns would fail in shear, thus showing an elastic-brittle behavior that would affect the overall behavior of the structure. In more detail, the shear demand of the column was evaluated by means of the capacity design starting from the moment-curvature relationship of the uppermost section (Figure 5). The classic bilinear stress-strain constitutive law with post-yield strain-hardening was adopted for the reinforcing steel bars (Figure 5a), whereas the classic parabola-rectangle stress-strain constitutive law with ultimate strain $\varepsilon_{cu} = 5$‰ was adopted for concrete (Figure 5b). The value of $\varepsilon_{cu} = 5$‰ was assumed for a three-fold reason, as described hereinafter. Firstly, to properly take into account the confinement effect yielded by the steel spiral, which yields an increase of the ultimate strain. Secondly, to take into account that, due to the presence of a circular cross section, failure is determined by the attainment of $\varepsilon_c = 3.5$‰, not in the outermost compressed fiber, but along a deeper one, which coincides with the centroid of the compressed area (Figure 5c). Lastly, assuming failure coincided with the achievement of the ultimate strain along the uppermost fiber, the horizontal tangent to the circular section would be excessively conservative.

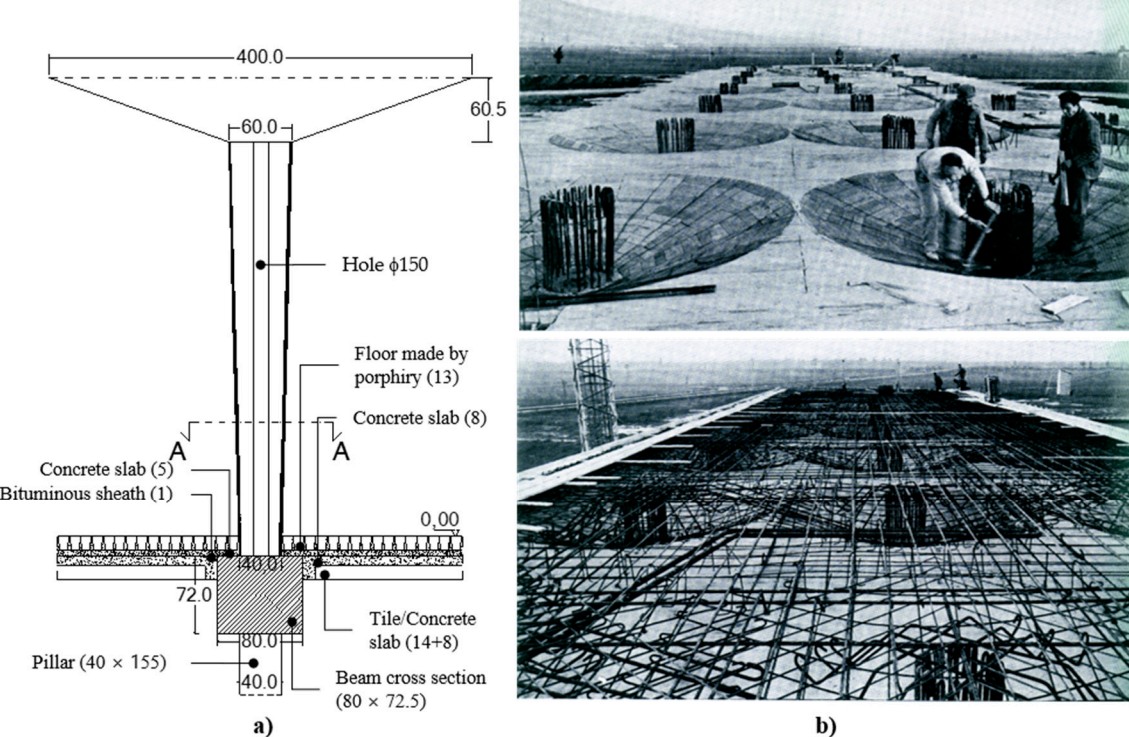

**Figure 3.** The peculiar mushroom-shaped columns of the ground floor: (**a**) vertical section, and (**b**) archive pictures of their prefabrication in situ (all dimensions in cm).

With the premises above, the moment-curvature $M$-$\phi$ relationship for the uppermost cross section of the column was evaluated by the classical hypothesis of planar deformation and by integrating the stress field resulting from the adopted constitutive laws (Figure 5d). The circular crown cross section has an external and internal diameter of 600 and 150 mm, respectively. The value of the axial load, resulting from the load combination corresponding to the service limit state, was equal to $N_{Sd} = 800$ kN. The yield point, singled out by the attainment of the yielding strain $\varepsilon_{yd} = f_{yd}/E_s$ along the most stretched reinforcing steel fiber, was characterized by curvature $\phi_{yd} = 0.00435$ m$^{-1}$ and $M_{yd} = 530$ kNm. The failure point of the $(M; \phi)$ curve, singled out by the attainment of the ultimate strain along the most compressed concrete fiber, was characterized by curvature $\phi_{ud} = 0.038$ m$^{-1}$ and bending moment $M_{ud} = 710$ kNm.

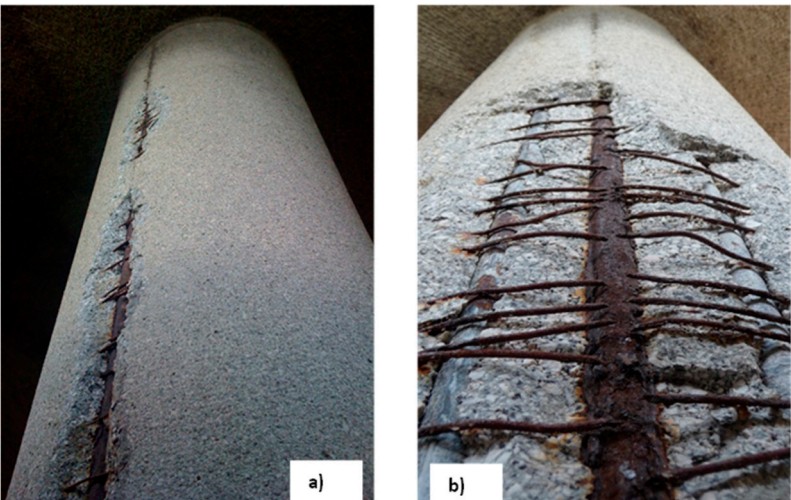

**Figure 4.** Details of the columns at the ground floor: (**a**) spalled concrete cover, and (**b**) corroded steel reinforcement.

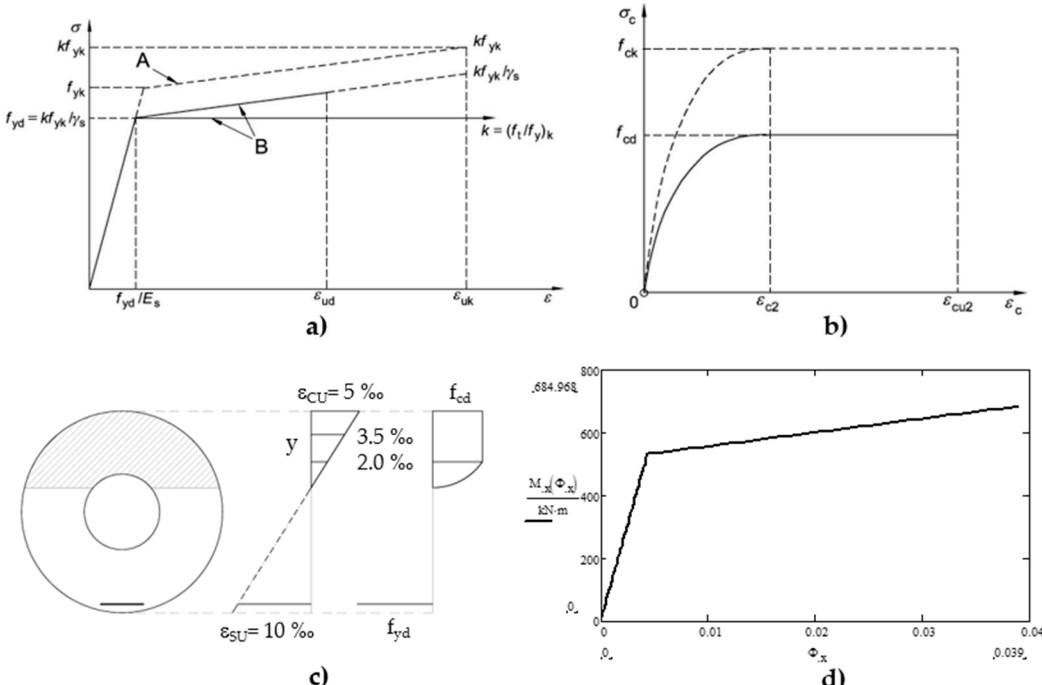

**Figure 5.** Some details of the safety evaluation: constitutive law of (**a**) reinforcing steel, and (**b**) confined concrete; (**c**) circular crown reinforced concrete cross-section strain and stress distribution; (**d**) resulting moment-curvature diagram.

The design value of shear demand $V_{Sd}$ at the lowermost cross section of the column was then evaluated according to the capacity-design philosophy, assuming that the top cross section could have already yielded, obtaining:

$$V_{Sd} = \frac{M_{yd,top} \times CF}{H} = \frac{636}{3.93} = 162 \text{ kN} \tag{1}$$

where $H$ is the height of the ground floor column, $CF = 1.2$ is the confidence factor, a function of the attained level of knowledge, and $M_{yd,top}$ is the design value of the yielding bending strength of the top cross section. The design shear capacity $V_{Rd}$ was evaluated by

means of the classical formulae of the Italian Standards for Construction [29], which are recalled hereinafter for the sake of clarity:

$$V_{Rd} = min(V_{Rsd}; V_{Rcd}) \tag{2}$$

$$V_{Rsd} = 0.9 \times d \times \frac{A_{sw}}{s} \times f_{yd} \times (cot\alpha + cot\theta) \times sin\alpha \tag{3}$$

$$V_{Rcd} = 0.9 \times d \times b_w \times \alpha_c \times f'_{cd} \times \frac{cot(\alpha) + cot(\theta)}{1 + cot^2(\theta)} \tag{4}$$

where $V_{Rsd}$ and $V_{Rcd}$ are the design values of the shear strength due to failure of transversal steel reinforcement and concrete strut, respectively; $\alpha$ and $\theta$ are the inclination angles of transverse steel and concrete compressed strut, respectively; $A_{sw}$ is the area of transverse steel reinforcement at $s$ given spacing; $f_{yd}$ and $f_{cd}$ are the design values of yield and compressive strength for reinforcing steel and concrete, respectively; $b_w$ is the width of the concrete cross-section web; $d$ is the distance of the centroid of the stretched bars from the most compressed concrete fiber. Using the equations above, for the current case and considering a reducing factor of 30% that takes into the account the degradation and corrosion of the concrete and the stirrups, the shear capacity of the base section, namely the least resistant, is:

$$V_{Rd} = min(V_{Rsd}; V_{Rcd}) = 69 \text{ kN} < V_{Sd} = 162 \text{ kN} \tag{5}$$

This result confirmed that the ground floor columns had a brittle behavior because shear failure was achieved in advance, with respect to the development of the flexural hinge at the top section.

Another weakness was represented by the presence of exiguous technical joints between the central edifice and each one of the lateral adjacent buildings. In fact, such gap, equal to approximately 40mm, was inadequate to properly accommodate the horizontal out-of-phase oscillations of the adjacent buildings during an earthquake. Such inadequacy would result in the occurrence of pounding (e.g., [27,28]), which is the mutual hammering that an adjacent building oscillating out-of-phase would undergo during an earthquake. Such phenomenon would be very dangerous from the point of view of the Life Safety Limit State (LSLS) since it might yield the collapse of non-structural elements, among which the in-fills in particular could fall ruinously to the ground on the escape routes, with possible tragic consequences for the occupants intending to leave the building during an earthquake. Such phenomenon may also yield significant change of the dynamic characteristics of the colliding buildings and structural damages. The necessary dimension of these technical joints was estimated to be equal to at least 100 mm on each side of the central building [30].

It should be noted that the joints are crossed by a system of optical fibers with which all data coming from the Italian highway network are sent in a protected underground area where they are recorded in a racks data center and processed.

The last technical intervention before starting the translation of the buildings was to prevent any interruption of data transmission, paying special attention to move all data lines that were able to interfere with the translation phase.

### 2.3. Design Constraints and Feasible Solution

As highlighted in the previous section, the seismic rehabilitation design had to start from fixing the two main sources of seismic vulnerability, which were found to be: (1) extremely low shear strength of the ground-floor mushroom columns, and (2) possible occurrence of pounding between adjacent buildings due to the insufficient dimension of the technical joints between them. At the same time, the owner of the buildings, together with technicians from Florence Superintendence for Cultural Heritage, imposed that any adopted design strategy should fulfill the following constraints:

- The work activities on the upper floors should not be interrupted during the construction site operations;
- The characteristic "mushroom" shape of the ground floor columns should not be modified, and therefore any radical strengthening interventions that would alter their shape and figurative value had to be excluded;
- The ground inter-story net height had to be kept unchanged;
- The hollow cross section of some of the columns, dedicated to housing the sewage ducts, had to be maintained.

The presence of a very stiff reinforced concrete box at the basement, together with the accessibility of the ground floor mushroom columns, suggested base-isolation as the most effective strategy for seismic rehabilitation. In fact, the presence of a rigid basement fulfilled the requirement of both Italian Standard for Constructions [29] and EuroCode 8 [31] to have a rigid diaphragm right below the so-called isolation layer, which is the horizontal layer containing the isolators (Figure 6).

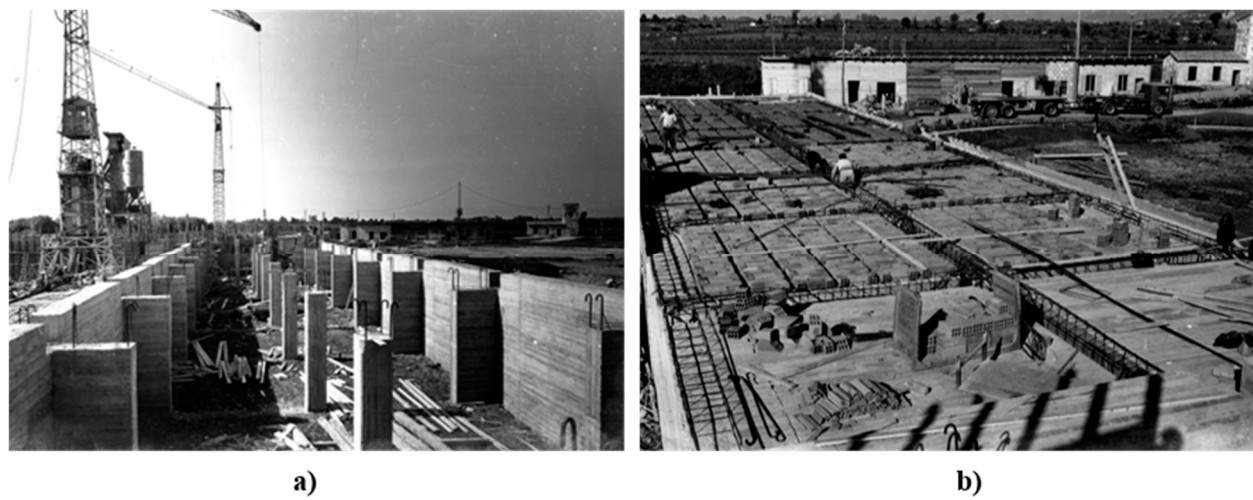

**Figure 6.** Archive photos of the Reinforced Concrete (RC) box at the basement: (**a**) before and (**b**) after the realization of the slab (note that the clay blocks, with which the slab is made, are clearly visible).

On the contrary, the expressed constraint not to alter either the shape or the net inter-story of the ground floor columns initially raised some concern due to the consequent impossibility of realizing a stiff base-mat above the isolation system in compliance with the same regulations. However, the regulations, imposing the necessity to have a rigid diaphragm both below and above the isolation layer, mean to guarantee that isolators work in unison. It was realized that in this case study, such a requirement would be fulfilled even in absence of a stiff-base mat at the extrados of the isolation layer since the seismic actions at the intrados of each isolation device would be synchronous and with the same amplitude. Such in-phase deformation of all the isolators would be guaranteed by a two-fold reasoning: (1) seismic action at each point of the foundation can be assumed synchronous due to the limited plan dimensions, and (2) seismic waves arrive at the installation quota of the isolators without any offset due to the filtering action of the very stiff basement underneath. Isolators would be placed right underneath each ground floor column. At the same time, for the problem of insufficient dimension of the technical joints, the horizontal translation of the two lateral buildings away from the central one, with consequent enlargement of the joints, seemed the most sustainable solution from both an economic and feasible standpoint. This strategy proved to be optimal from all standpoints—technical, architectural, functional—and allowed for attaining seismic rehabilitation without interfering with the activities carried out inside the building, keeping the office workers completely safe during all construction phases.

After deciding that the strategy would integrate base isolation to the building translation, a feasibility study was carried out in order to single out the technical solution that presented the most interesting ratio between advantages and disadvantages. The Lead Rubber Bearing (LRB) Isolators were selected due to the presence of a lead core, which provides the isolators with a larger local damping.

During the preliminary feasibility study, several ideas arose that are worth reporting in view of further application and/or refinement of the overall strategy, listing both advantages and disadvantages. As to the lifting operations, propaedeutic to cut the ground-columns and insert the isolators, the first proposal envisaged the presence of two temporary steel columns (Figure 7a) placed along a diameter of the column cross section and on opposite sides to support the two hydraulic jacks. This solution presented the advantage of a reduced number of bearing points on the ground floor slab but was discarded for the difficulty of keeping the verticality of the loading jacks. As the system was to translate the building, the initial idea contemplated the preventive installation of a bearing plate, lined with a leaf of PolyTethraFluoroEthilene (PTFE) on which to slide the LRB isolator (Figure 7b), by means of a horizontal hydraulic jack. Since the jacks would apply the hydrostatic force on the columns of one side only, a temporary steel horizontal frame connecting the base sections of the columns would complete this solution. However, this solution was successively discarded since it presented many drawbacks, among which: (a) difficulty to fix the sliding bearing lined with PTFE on the ground floor deck for the presence of brittle clay bricks; (b) difficulty to provide a suitable contrast structure for the horizontal hydraulic jacks during the translation of the building; (c) expensiveness of the temporary steel bracing structure. As for the LRB isolators, one of the early ideas for a device to be placed beneath the hollow columns was a drilled isolator (Figure 7c). This solution presented some advantages, among which: (a) ability to house the sewage duct, and (b) better numerical control since it is completely elastic due to the absence of a lead core. Among the disadvantages: (a) excessive vertical deformability due to the reduced cross section with respect to the intact isolators installed close by; (b) concentration of stresses on the horizontal slab at the upper floor due to the different deformability of close isolators; (c) lateral buckling of the rubber circular crown for displacements of about 100 mm; (d) p-delta effects during the earthquake and during the re-centering. For the former point, note that, despite the presence of an internal hole, it was not possible to hypothesize a larger outside diameter in order to have an overall transversal area equivalent to the not-drilled isolators, since the Superintendence technicians imposed that the plan cross section of the isolators did not overcome the base cross section of the ground columns.

The technical and economic feasibility study was concluded with a very brilliant proposal, which consisted of integrating base-isolation and horizontal translation of the lateral buildings away from the central one. In fact, the best solution, from a technical–economic standpoint, consisted of mounting in place LRB isolators that were already deformed, with a horizontal displacement exactly equal to the quantity of which each building had to be translated, i.e.,100 mm. Underneath the hollow columns, special devices, designed ad hoc and named tripods, would be placed. Such tripods would be composed of three horizontal sliders each, placed in the vertices of an equilateral triangle inscribed in a circle in plan whose center was located in the center of the base cross section of the hollow column. In this way, in the central part, amid the three horizontal frictionless sliders, there would result a hollow zone in which the sewage duct coming from the central part of the column above could be housed. Both LRB isolators and tripods would be put in place already deformed. The adopted solution would be completed by (1) a system of hydraulic jacks to thrust the lateral buildings in the new equilibrium position, and (2) a gradual release system, displacement controlled, necessary to reduce the disturbance brought to the occupants during the building translation. The adopted technical solution is presented with increasing levels of detail in the next section.

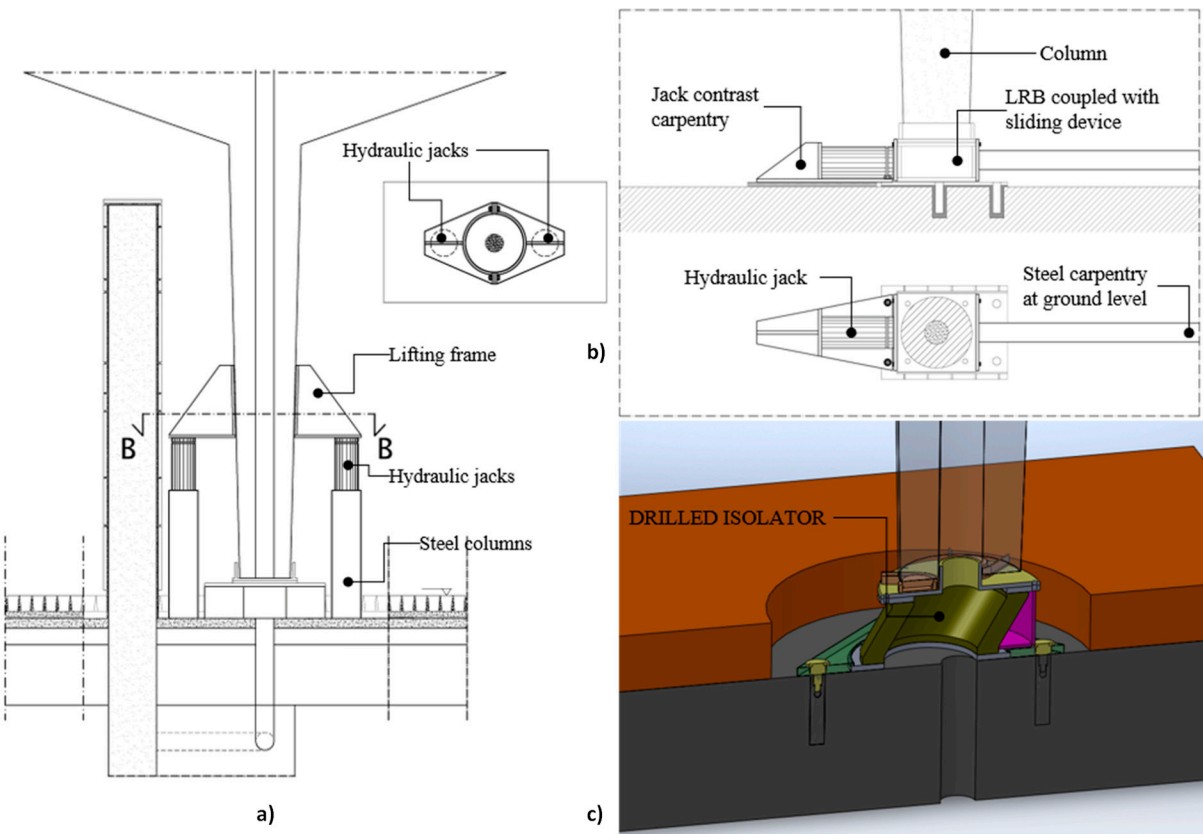

**Figure 7.** Feasibility study and first proposals of technical solutions: (**a**) lifting system to cut the columns, (**b**) thrusting system, (**c**) drilled Lead Rubber Bearing (LRB) isolator to house the sewage duct.

## 3. Seismic Rehabilitation Design

The building before rehabilitation shows the seismic adequacy level, expressed as the ratio between Capacity and Demand in term of shear, equal to 0.45.

With the intervention of the base-isolation and shear strengthening of the columns of the ground floor (to respect the capacity design criterion), the building goes up to a safety factor of 1.21.

### 3.1. Shear Strengthening of Ground-Floor Columns by CAM®

Even though the global strategy of base isolation has greatly reduced the shear in the columns, the shear demand is still greater than the one taken by the pre-existing columns with steel spiral-shaped wire.

For this reason, the CAM® technique was chosen, where CAM is the acronym for the Italian expression "Cucitura Attiva dei Manufatti" (e.g., [32,33]). This technique was chosen since it guaranteed rapidity of execution and minimum change of volume, with consequent better preservation of both shape and dimensions of the prefabricated columns. This technique consists in the employment, instead of the classical steel stirrups, of stainless high strength steel strips that are pretensioned and successively fastened with special clips. In this way, in the most general cases, they also exert a triaxial compression that confines the concrete, thus improving its compressive strength. The mechanical characteristics of a single strip are listed in Table 1, while the application phases and the result are shown in Figure 8. The design strength of the CAM strips $f_{yd, CAM}$ is evaluated as the minimum of two contributions, which are (1) the one ascribed to the yield strength of a cross section

far from the clip, and (2) the one due to the net cross section in correspondence with the fastener, as follows:

$$f_{yd,CAM} = min\left\{\frac{f_{yk}}{\gamma_{M0}}; \frac{0.7 \times f_{tk}}{\gamma_{M2}}\right\} = 532 \text{ MPa} \tag{6}$$

where $f_{yk}$ and $f_{tk}$ are the characteristic values of the yield and tensile strength of the stainless high strength steel, respectively; $\gamma_{M0} = 1.05$ and $\gamma_{M2} = 1.25$ are the two safety factors. The relevant value of the design shear strength $V_{Rsd}$ of the strengthened cross section of the column can be evaluated by means of Equation (3), in which $A_{sw}$ is assumed equal to twice the strip cross section $A_{sw} = 2 \times t_s \times w_s$ and the design yield strength is assumed equal to $f_{yd,CAM}$. The case study columns were strengthened by means of one strip every 90 mm, attaining for a final strength of $V_{Rsd} = 197$ kN.

**Table 1.** Mechanical properties of CAM® stainless-steel strips.

| Property | Symbol (Unit) | Characteristic Value |
|---|---|---|
| Thickness | $s$ (mm) | 0.9–1.0 |
| Width | $b$ (mm) | 19 |
| Yield strength | $f_{yk}$ (N/mm²) | ≥840 |
| Ultimate tensile strength | $f_{tk}$ (N/mm²) | ≥950 |
| Ultimate elongation | $(A_{gt})_k$ (%) | ≥20 |

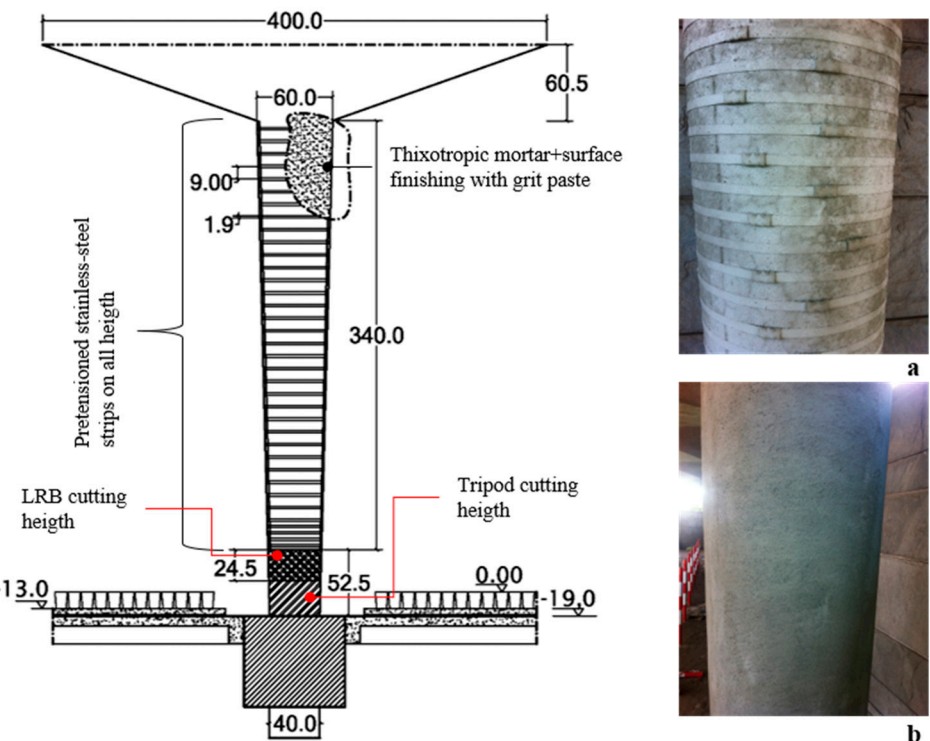

**Figure 8.** Preliminary shear strengthening of the ground floor RC columns: (**a**) pretensioned strips after applying a gripping layer, and (**b**) view of the strengthened column after applying the plaster (all dimensions in cm).

## 3.2. Lifting Frames

The lifting system consisted of a steel frame placed around the column to be cut, composed of three or four columns and horizontal bracing elements (Figure 9). These horizontal frames were intensified in the upper part of the frame in correspondence with the placement of the hydraulic jacks, which exert the thrust leaning against the inclined

surface of the mushroom columns, in order to absorb the inclined reaction (Figure 10). The three-columned steel frames were employed for the columns that are close to the vertical RC shells, delimiting the walkable area of the ground floor as one side was thus not accessible. These temporary supporting frames were realized by means of elements of manageable dimensions and equipped with handles which could be easily transported and joined in situ. These elements were joined in situ by means of bolted plates and could be both re-utilized and adapted to other case studies. The adopted steel frames were of galvanized steel with a grade of S375 [29], of which the columns were realized by means of hollow square sections of dimensions 200 × 200 × 8 mm and the horizontal bracing frames were realized by means of L-shaped open profiles of dimensions 160 × 160 × 8 mm. The employed hydraulic jacks had a 300 kN capacity each. The unloading of the column was made by applying four (or three) concentrated loads upwards of 200 kN each (270 kN) for a total value of 800 kN, which is the axial force loading the column for the minimum static load combinations [29]. In particular, the four (or three) hydraulic jacks placed around the column were activated simultaneously by a control system and were halted at the attainment of either one of the following conditions: (1) the total applied force reached the maximum value of 800 kN (710 kN), or (2) the elongation of the column reached the value of 0.3 mm.

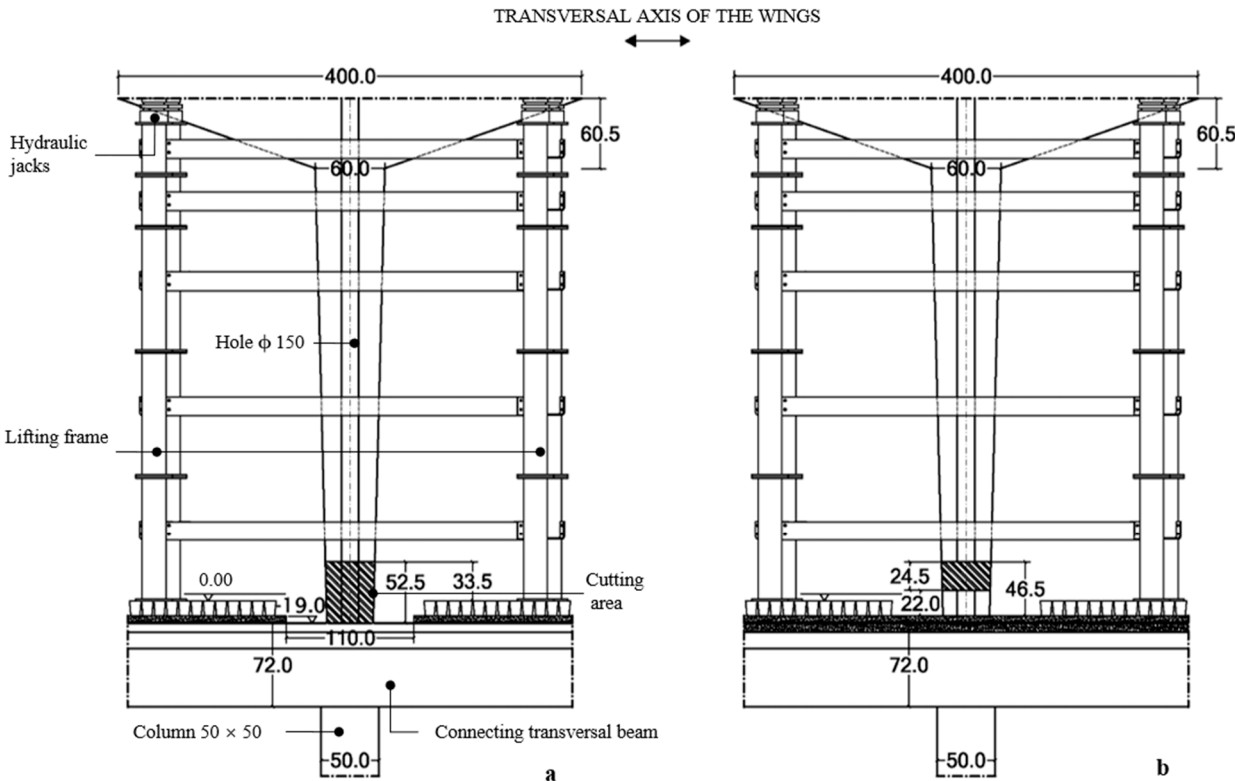

**Figure 9.** Placement of the steel lifting frame and the cutting zone: (**a**) for the tripod and (**b**) for the LRB device (all dimensions in cm).

After unloading the column as specified above, the cut was carried out by means of a diamond-head saw (Figure 11). It was important, at the time of lifting the column, to check its deformation so as to avoid residual tensile or compressive stress. In fact, if this latter did not vanish, the cutting operation was difficult, if not impossible, due to the "braking effect" induced by the axial load on the advancement of the diamond-head saw.

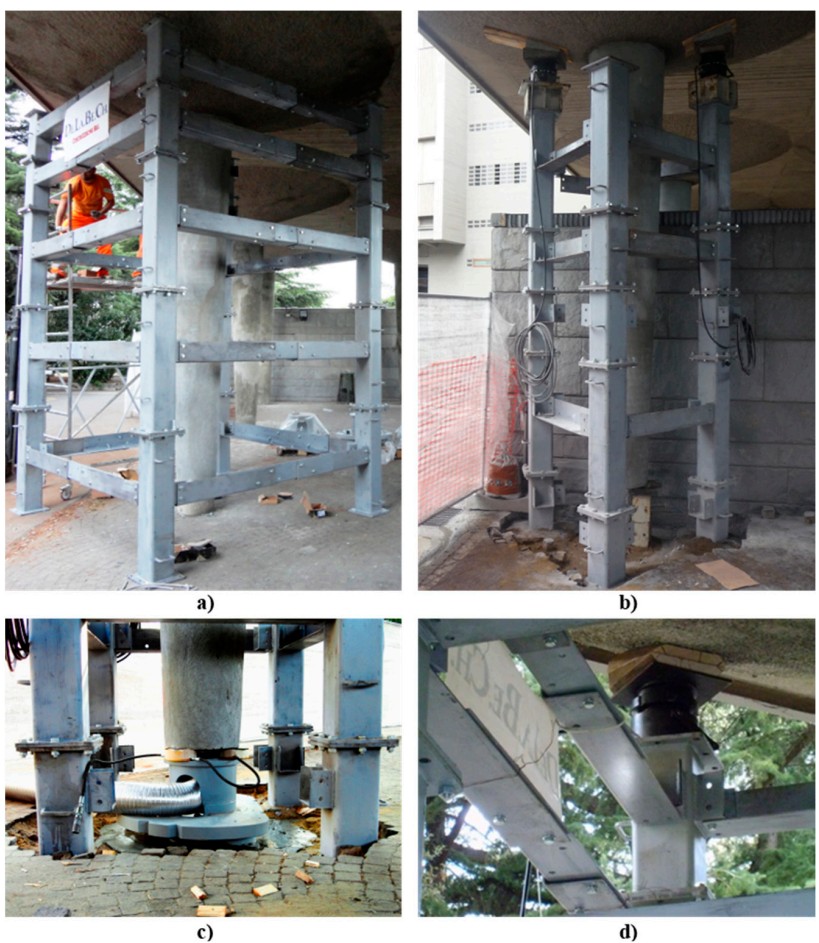

**Figure 10.** Different lifting steel frames: (**a**,**b**) the four- and three-columned ones, respectively, (**c**) base detail of the lifting frame, and (**d**) adjustment of the hydraulic jack head on the inclined surface of the mushroom column.

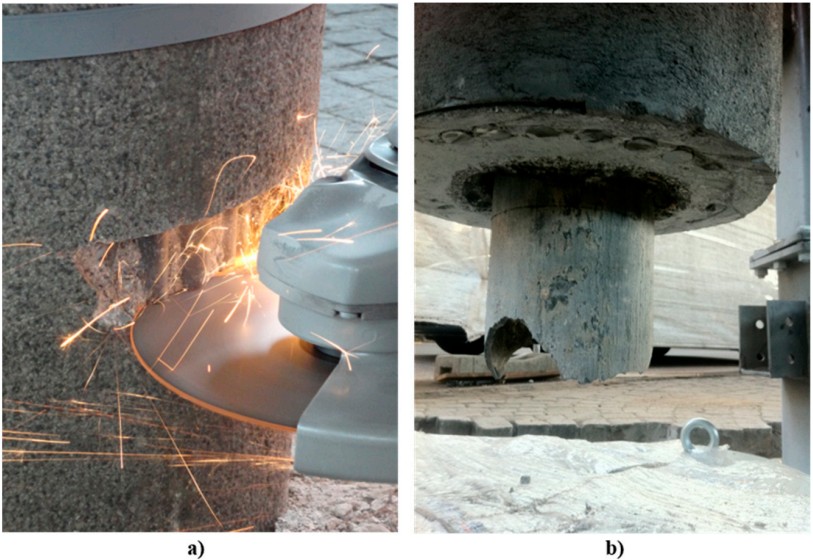

**Figure 11.** Cutting operations of the ground floor columns: (**a**) cut of the existing longitudinal reinforcing steel bars by a diamond-headed saw, and (**b**) hollow column with sewage duct after cutting.

### 3.3. Base Isolation System by Pre-Deformed Lead Rubber Bearings

The base isolation system was designed in order to obtain an optimal distribution of the devices with no torsional effects. The total stiffness of the devices was calibrated so that the building's fundamental vibration period would shift to the zone of the pseudo-acceleration spectrum, characterized by lower values of inertial forces and larger displacements. The target period was $T_1 = 2.4$ s, which corresponds to a displacement of 100 mm.

Two devices were used in the isolation system design: (1) the lead rubber bearing devices characterized by a high level of energy dissipation capability with wide and stable force-displacement hysteresis loops; (2) sliding devices linked in-parallel, named "tripods," to follow the seismic displacements without interfering with the dynamic response of the isolated system yielded by the LRB devices.

As for the LRB devices, the hybrid combination of rubber and lead can in general guarantee a high level of seismic energy dissipation, with a value of equivalent viscous damping ratio up to 30% [34]. In this case, the equivalent viscous damping ratio assumed for the adopted LRB devices was equal to 16%, since the lead core of the adopted LRB specimens was smaller than the average and the seismic design displacement was not so large.

As for the complementary sliding devices, whose behavior is rigid and perfectly plastic, they contribute with a force obtained by multiplying the friction coefficient present at the sliding interface by the axial load at the service limit state. The sliding interface is made by coupling stainless steel and PolyTetraFluoroEthylene (PTFE), resulting in a friction coefficient of approximately 1.5%.

Both LRBs and sliding devices had to satisfy the following functions:

- allowing the movement of the superstructure in seismic conditions for safeguarding the integrity of both the carried structure itself and its contents, including both occupants and non-structural elements;
- supporting the axial load transferred after removing the relevant lifting frame;
- facilitating the thrusting operations necessary to horizontally translate the lateral wings in their new position;
- housing the sewage ducts contained within the relevant column and allowing for their maintenance.

As for the third function, the lateral wings had to be translated outwards, from the central building, about 100 mm in order to reach a position in which any possible out-of-phase oscillation of the adjacent edifices would yield no pounding. The fourth function was specific to the tripods, which will be shown more in detail hereinafter. They were especially designed to allow the passage of the sewage ducts.

A special removable case made with galvanized metal was used to protect each device, both LRBs and sliders, after their installation. Such a protective case was necessary in order to protect the devices against any source of damage. In fact, particular attention had to be paid to the protection of the LRBs against the ultraviolet rays, which may have induced the splitting of the polymer chains of the natural rubber constituting the isolators, causing an abrupt decay of their physical and mechanical properties. The protective case was also useful in avoiding any penetration of dust or dirt on the sliding surfaces of the tripods. In fact, the accumulation of dirt must be avoided in order to maintain the frictional properties assumed during the design. At the same time, the protective case had to resist the passage of both pedestrians and cars. This strategy proved to be optimal from all standpoints—technical, architectural, functional—and allowed the fulfillment of the requests of the Superintendence. The adoption of removable protective cases also warranted the inspection and substitution of damaged devices. The final result is also pleasant from an architectural point of view, as can be gathered from Figure 12.

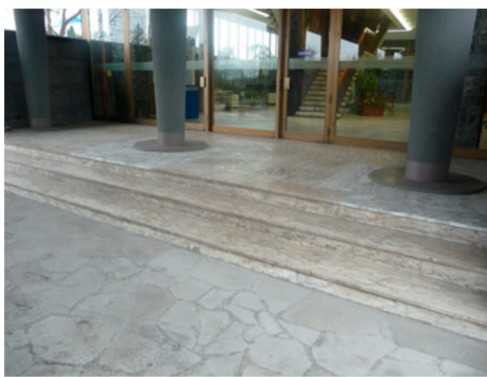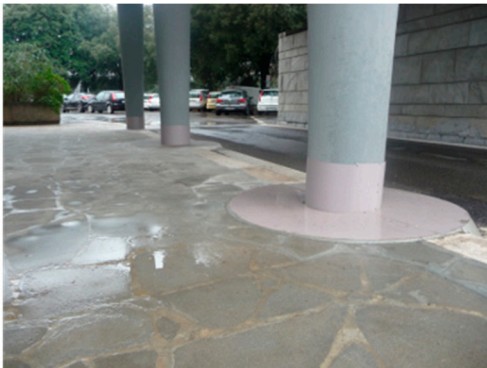

**Figure 12.** Visual impact of the intervention at the end of the work.

### 3.4. Pre-Deformed Lead Rubber Bearings

The adopted LRB prototype isolator is represented in Figure 12. The whole package of alternate thin layers of Natural Rubber (NR) and steel presents the shape of a straight cylinder with circular horizontal cross sections. The height of this cylinder is 1760 mm, and the diameter of the circular cross section is 350 mm long. The diameter of the internal lead core is equal to 54 mm. Thus, the lead to rubber area ratio is equal to 15.43%. This ratio corresponds to an equivalent viscous damping ratio of $\xi = 16$ %. Taking into account this added damping, by means of the corresponding value of the coefficient $\eta = 0.7$ [29], the pseudo-acceleration corresponding to the Life Safety Limit State (LSLS) was reduced to $S_{A,e} = 0.06$ g by increasing the vibration period to $T_1 = 2.4$ s. The corresponding value of spectral displacement demand amounted to $S_D = 90$ mm. With this significant reduction of the acceleration demand, it was possible to dramatically reduce the strengthening interventions on the RC structure, thus fulfilling the minimum intervention explicitly requested by the Superintendence.

The device was designed, in compliance with the Italian regulations [29], for a target displacement of 100 mm, which corresponds to the Collapse Prevention Limit State (CPLS) design for earthquakes. Several components, made of NR, stainless steel, and lead, were designed in compliance with the harmonized European Standards [35,36]. The physical and mechanical properties of the adopted materials are listed in Table 2. Note that the device was endowed with two co-axial short steel tubes, both at the extrados and at the intrados, which were necessary for its installation in place.

In this way, the mechanical characteristics of the LRB devices and the complementary sliding devices were defined in order to seismically rehabilitate each of the lateral edifices, if these were far enough from the central building, so as not to interact with it. However, as already stated, the gap between each of the lateral edifices and the central building was already insufficient in the ante-operam conditions. It would be even more insufficient after implementing the base isolation due to the increment of the displacements undergone by the superstructure during the design for earthquakes.

For these reasons, it was decided to seize upon the intrinsic deformability of the LRB devices and their re-centering capability in order to integrate the needs to base isolate the buildings with that of horizontal translation. Thus, the decision was made to pre-deform the LRB isolators according to the maximum displacement capacity foreseen during the design, which amounted to 10 cm, and to install them in situ with the lower plate in the position in which the building had to be translated, and the upper plate in the current position of the building. In this way, we would seize upon the re-centering action of the devices and drag each lateral edifice in its new position, 100 mm farther from the central building.

The LRB devices were provided by the production factory already deformed and blocked in such positions by means of removable steel constraints, which rigidly connected the lower and upper plate (Figure 13).

Table 2. Physical and mechanical properties of the Natural Rubber (NR).

| Physical/Mechanical Property | Value | Unit | Limit Value | Reference Code |
|---|---|---|---|---|
| Nominal hardness | 57.3 | Micro IRHD | 60 | ISO 48:2010 |
| Hardness range | | Micro IRHD | 56–65 | |
| Tolerance hardness | | Micro IRHD | ±3 | |
| Ultimate strength | 21 | N/mm$^2$ | ≥16 | ISO 37:2005 |
| Ultimate elongation | 671 | % | ≥425 | ISO 37:2005 |
| Plastic deformation | 24.2 | % | ≤ 30 | ISO 815-1:2008 |
| Max. variation of hardness | +1.9 | Micro IRHD | − 5–+10 | ISO 188:2007 |
| Max. variation of ultimate strength | +6.7 | % | ±15 | ISO 188:2007 |
| Maximum variation of ultimate elongation | −6.5 | % | ±25 | ISO 188:2007 |
| Ozone-strength | No visible crack 7X | | Not cracked | ISO 1431-1:2009 |
| G module | | N/mm$^2$ | 0.8 | |

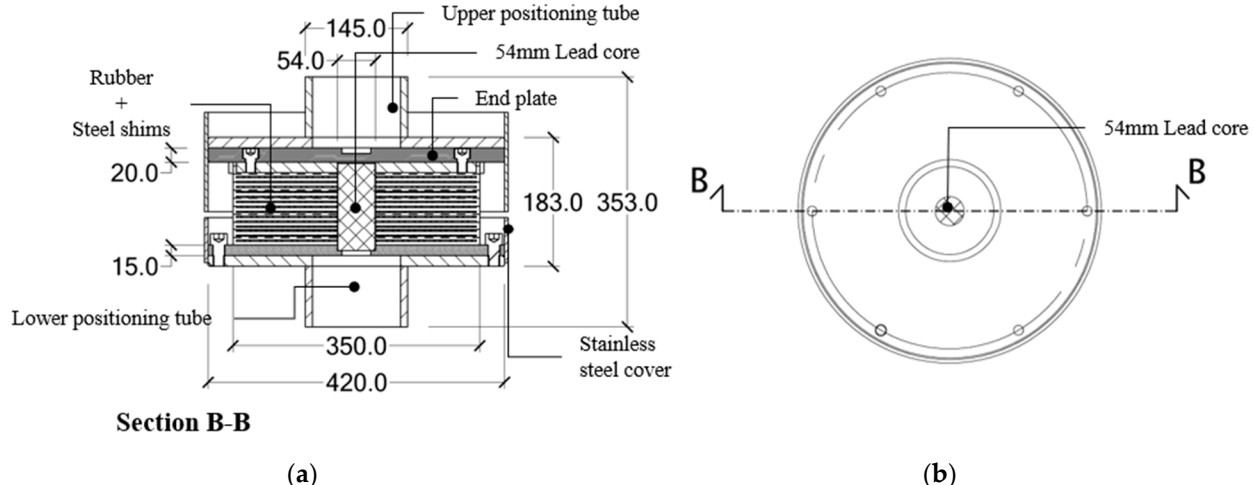

**Figure 13.** The adopted LRB isolator: (**a**) vertical section, and (**b**) horizontal plan view (all dimensions in mm).

The realization of the intervention foresaw the phases described hereinafter. The pre-deformed LRB devices were firstly stockpiled in the construction yard (Figure 14a). The ground columns were saw-cut, leaving an additional space at the extrados of the position of the isolators which was necessary to house the three hydraulic jacks adopted to load the column. The LRB was put in place, with its intrados steel plate fixed at the base thick RC slab, in the new position of the building, and with the extrados steel plate in the current position of the building and consequently of the columns (Figure 14b). Successively the three hydraulic jacks were positioned at the vertices of an equilateral plan triangle in such a way to exert their loading action against the circular crown constituting the column resisting cross section. In order to allow the removal and the recovery of the jacks, after loading the column each jack was protected by surrounding it with a U-shaped steel plate (formwork) to be left in place (shown in detail in the Figure 15). Successively the three jacks were loaded up to the complete unloading of the relevant lifting frame and the consequent transfer of the axial load above the LRB device. The resulting space between the upper co-axial positioning steel tube and the U-shaped plates was in-filled with a shrinkage-compensated cementitious mortar. After this latter had been cured, the three jacks were unloaded and recovered. Then the resulting spaces, initially occupied by the latter, were in-filled with the same mortar. After the removal of the hydraulic jacks and completion of each column, all the lifting frames were also removed. At that point, a precision displacement transducer was placed in correspondence of each LRB device in order to monitor their deformation during the horizontal translation of the buildings (Figure 14c). The temporary constraints were removed (Figure 14d) and each lateral edifice was thrusted in its new position (Figure 14e,f), by means of horizontal jacks placed in

contrast with the central building and in correspondence with the thick RC slab of the first floor. The thrusting operations were displacement-controlled by means of a system of cables at gradual release, described in a following paragraph, in order to limit as much as possible, the disturbance brought to the occupants and the working activities.

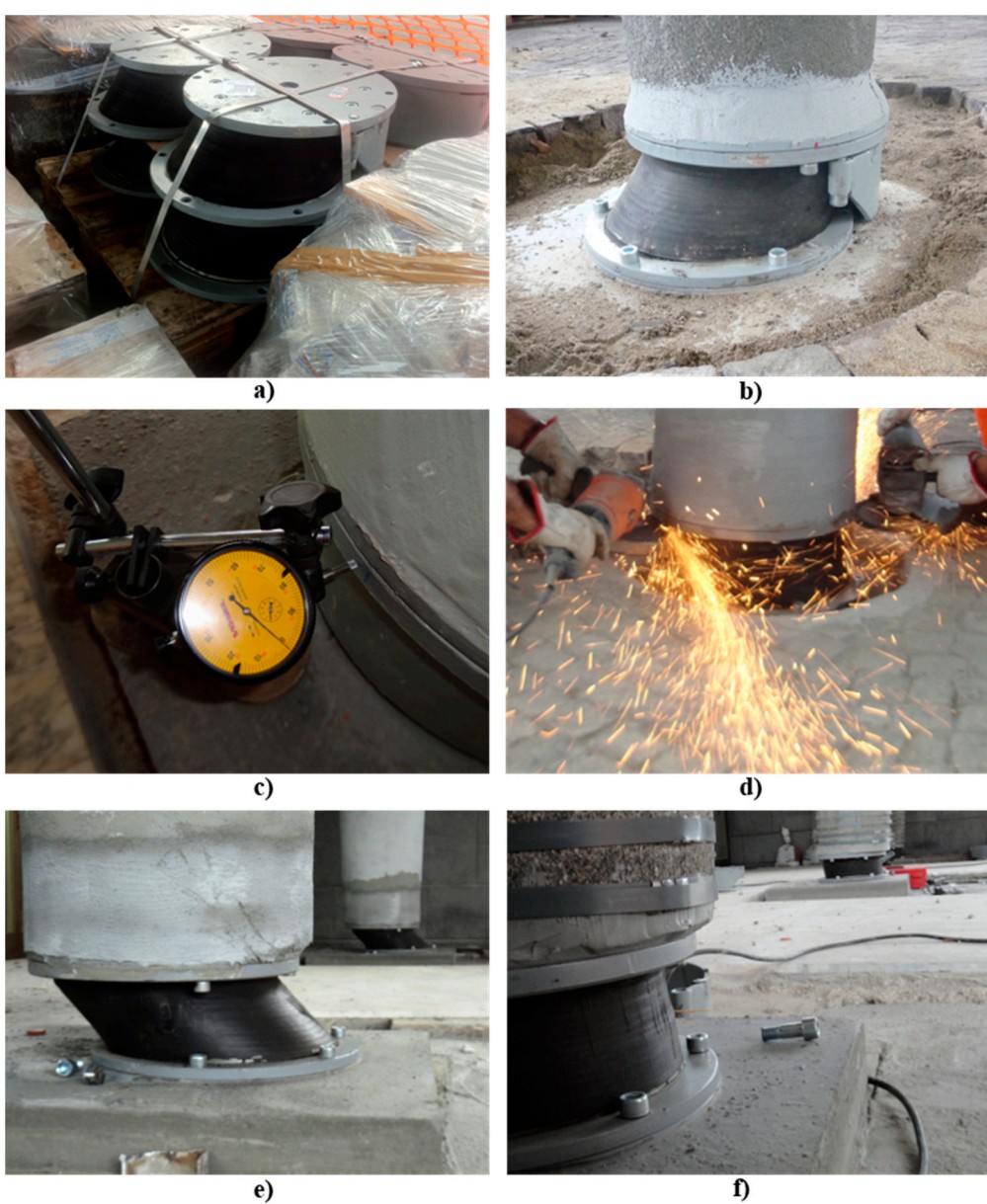

**Figure 14.** Some pictures of the pre-deformed LRB isolators: (**a**) storage of the devices at the construction yard; (**b**) pre-deformed LRB installed under the column; (**c**) installation of the high precision displacement transducer; (**d**) removal of the temporary constraint; LRB device at (**e**) the beginning, and at (**f**) the end of the re-centering operations.

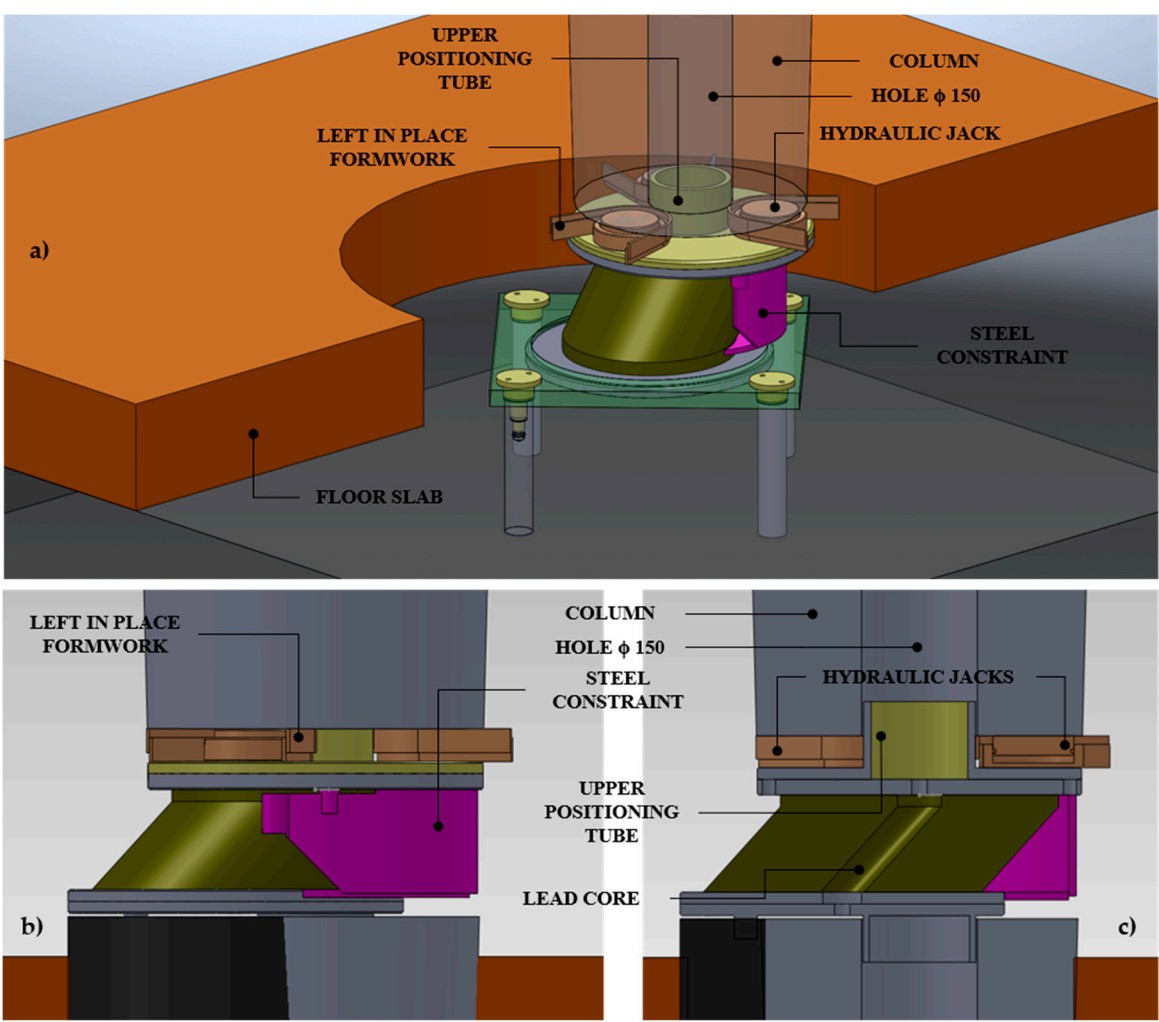

**Figure 15.** Some technical drawings (renderings) of the pre-deformed LRB isolator: (**a**) axonometric view; (**b**) lateral view; (**c**) vertical cross section along the diameter corresponding with the direction of translation.

In the following Table 3, the mechanical parameters on which the sliding properties of isolation system depend are listed.

### 3.5. Designed Tripods

Some sliding devices, labelled tripods, complemented the base-isolation system, which were composed of the pre-deformed LRBs described in the previous paragraph. These tripods were placed beneath the columns in which the sewage ducts, descending from the upper stories, were placed. These devices were designed in compliance with both the Italian regulations [29] and the harmonized European technical standards [35,36]. Each device is composed of a sliding plane, realized by coupling a horizontal surface in stainless steel and a horizontal surface lined with PTFE. Two parts compose the device: one at the intrados and another at the extrados of the sliding plane. The former is fixed to the underlying thick RC slab of the ground floor, while the latter is fixed, through its extrados, at the column above. The lower part is composed, proceeding from the underlying RC slab, of a 40 mm-thick steel plate and three pot-bearings. The steel plate, which is fixed at the RC slab, has a triangular shape in plan and presents an opening at the center. In each of the vertices of this plate, a pot-bearing is placed. Each of these pot-bearings is composed of a lower steel hollow cylinder, in which a thick layer of unreinforced elastomer is confined. The upper part of the pot bearing is completed by a steel piston, which pushes the elastomer downward as a function of the applied axial load. The elastomer, which

is subject to a triaxial pressure, behaves like a pressurized viscous fluid. The extrados of the steel piston presents a recess in which a layer of PTFE is placed. The elastomer allows any rotation with respect to a horizontal axis with whatever orientation is in the plan. The 10 mm-thick lowermost steel plate, which is fixed directly on the RC slab of the ground floor by means of steel bolts, presents a central hole to allow the passage of the sewage duct.

**Table 3.** Physical and mechanical properties of LRB (rubber only), lead, and stainless steel-PolyTetraFluoroEthylene (PTFE) interface.

| Physical/Mechanical Property | Natural Rubber (NR) | Lead | Stainless Steel-PTFE Interface |
|---|---|---|---|
| Static tangent modulus ($G_s$, Mpa) | 0.77 | 5000 | - |
| Dynamic tangent modulus ($G_d$, Mpa) | 0.85 | - | - |
| Static tangent strength ($\tau_s$, Mpa) | - | 5.95 (6–8) | - |
| Static tangent strength ($\tau_d$, Mpa) | - | 10.5 | - |
| Friction coefficient ($\mu$, %) | - | - | 1–1.5 |
| Height of Rubber ($T_e$, mm) | 75 | - | - |
| Area of Rubber ($A_r$, mm$^2$) | 93.873,4 | - | - |
| Area of Lead ($A_l$, mm$^2$) | 2.289,1 | 2290 | - |

The upper part of the tripod is composed of the parts described hereinafter, starting from the sliding plane and moving upwards. There is a 40 mm-thick horizontal steel plate with a central hole that presents at the intrados, in correspondence with each pot-bearing, a circular recess in which a layer of stainless steel is welded. This latter, together with the PTFE at the extrados of the underlying pot-bearing, concurs to form the sliding kinematic pair. At the extrados of this thick steel plate, a hollow vertical steel cylinder is welded. This cylinder presents an opening on the lateral surface in order to house the flexible part of the sewage duct, which has to accommodate the seismic horizontal relative displacements. The upper base of the cylinder is composed of a 20 mm thick steel plate, in the shape of a circular crown, coaxial to the column above (Figures 16 and 17).

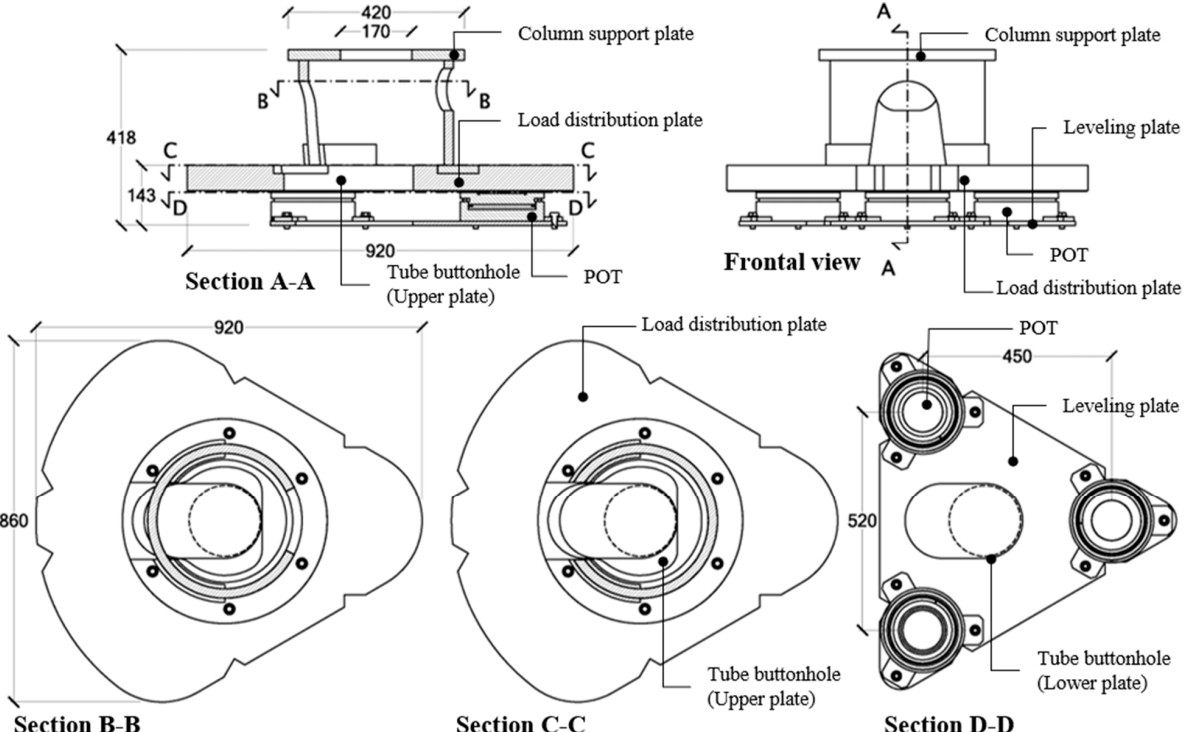

**Figure 16.** Technical drawings of the tripod, made during their design (all dimensions in mm).

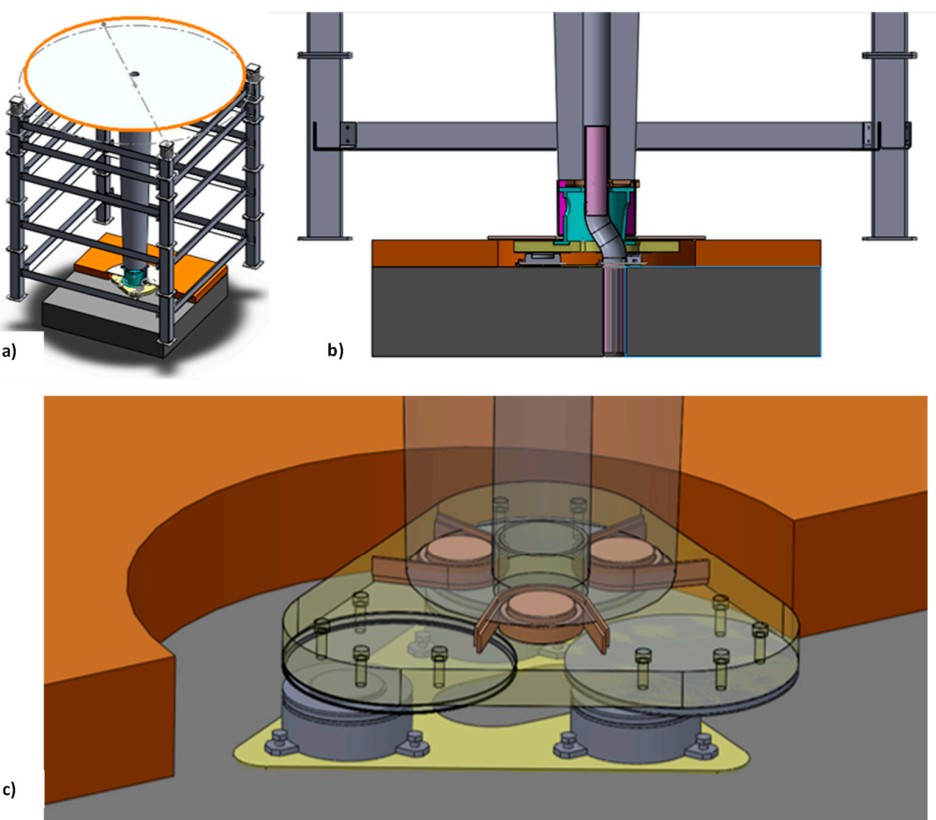

**Figure 17.** Technical drawings of the tripod made during their conception/design: (**a**) axonometric rendering of a whole mushroom-shaped column, (**b**) vertical cross section, and (**c**) close-up axonometric rendering of a tripod.

The transfer of the column's axial load from the temporary lifting frame to the tripod was carried out by means of three hydraulic jacks and the subsequent employment of a shrinkage-compensated cementitious mortar, as already described in the previous paragraph for the pre-deformed LRBs.

The adopted pot-bearings were acquired directly from a producer. On the contrary, the upper part of the tripod was purposely designed for this specific application. A 3D Finite Element Model was assembled, and numerical analyses were carried out for three load combinations, namely: (a) the Ultimate Limit State (ULS) for the gravity loads combination, (b) the seismic load combination for the CPLS, and (c) assuming ($b_1$, $b_2$) two directions in place for the main earthquake direction (Figure 18).

A picture of a typical tripod installed in place is represented in Figure 19.

### 3.6. Displacement-Controlled Re-Centering of the Lateral Edifices

The need to guarantee the temporal continuity of the working activity carried out inside the building and the safety of its occupants, during the operations of horizontal translation of the lateral edifices, lead to the design of a gradual release system to control displacements. Such a system foresaw the employment of two cables, each composed of two Dywidag® bars [37] made of steel grade S355 [29], arranged in plan as indicated in Figure 20. Each cable was fixed at its extremities at the intrados of the thick RC slab constituting the first floor by means of two steel anchorage elements, designed ad hoc and bolted in the slab's reinforced concrete. Corresponding with one of these two anchorages (the passive one), the cable was fully constrained while on the other one (active) was constrained by means of a hydraulic jack, which applied a pulling force. The overall value of the latter, evaluated excluding any non-linearity such as lead core's yielding, friction at the sliding interface of the tripods, and so on, was equal to $F_T = 1500$ kN.

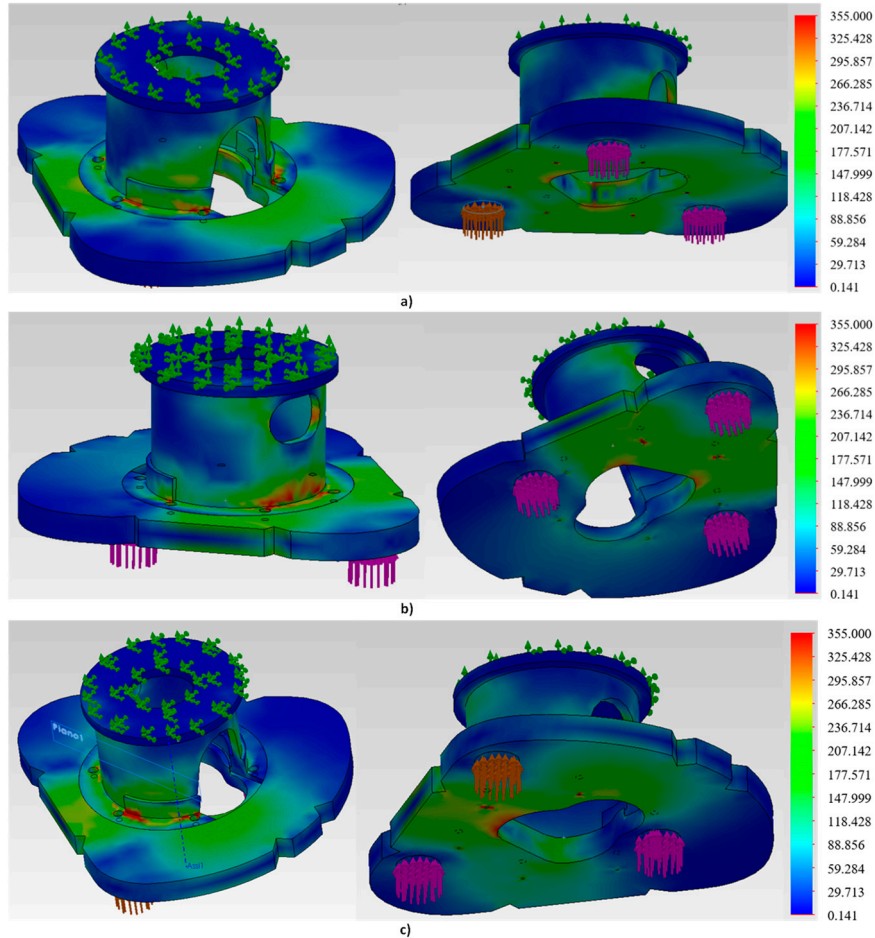

**Figure 18.** Finite Elements Model adopted to design the tripod: stress-field (N/mm$^2$) according to the Von Mises criterion, for (**a**) static loading combination a, and for seismic loading combination (**b**) $b_1$, and (**c**) $b_2$.

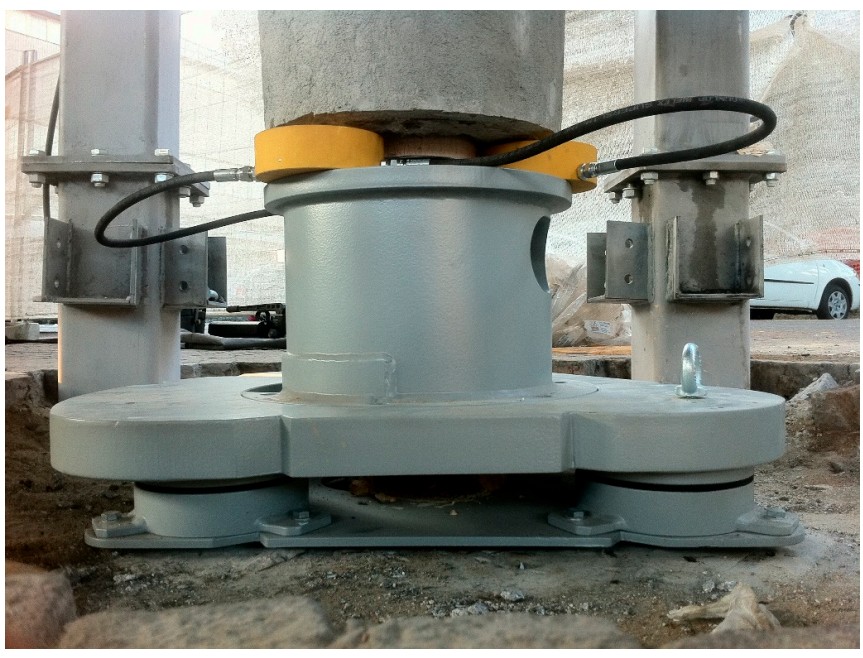

**Figure 19.** Picture of one of the tripods during its loading by means of three hydraulic jacks.

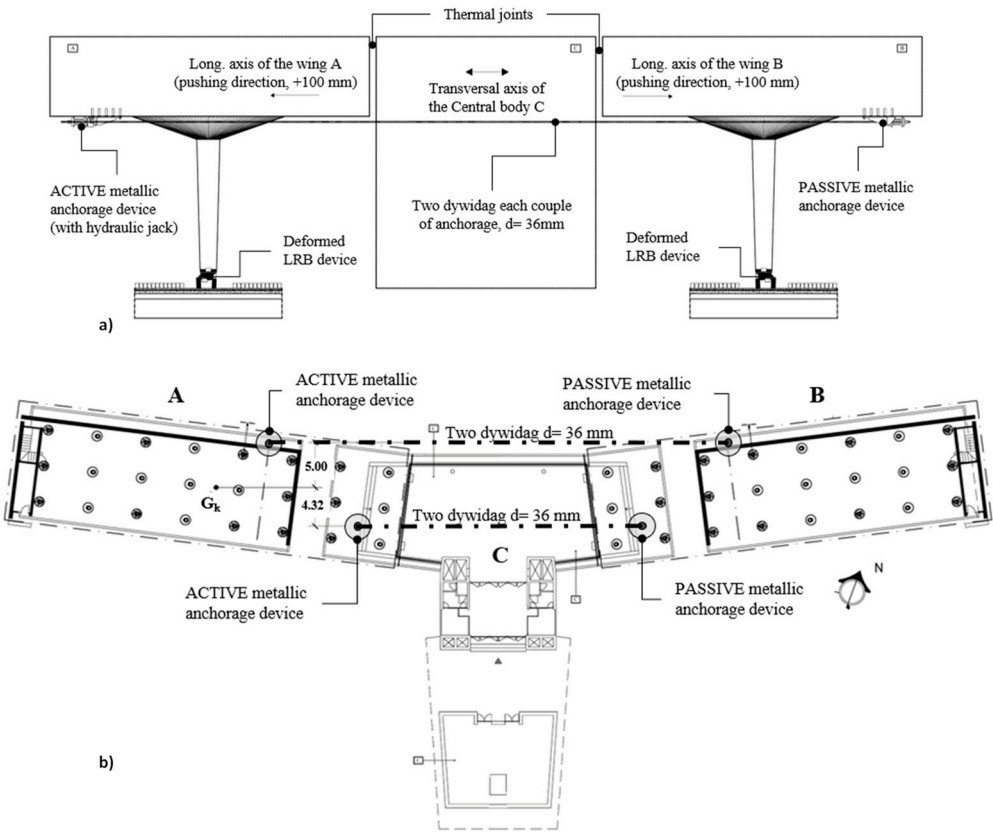

**Figure 20.** Schematic view of the system of cables at gradual release and the relevant components: (**a**) vertical section, and (**b**) plan view (all dimensions in m).

One of the two cables was internal and the other was external to the plan of the building (Figure 20b). The internal cable's pulling force and the distance of the relevant line of application from the line passing through the two centers of stiffness of the LRBs placed in the two lateral edifices (Figure 20b) are labelled as $F_{T,i}$ and $d_i$, respectively. Likewise, the same quantities concerning the external cable are $F_{T,e}$ and $d_e$. The pulling force to be applied by means of the relevant hydraulic jack in each cable was determined by imposing the eccentricity of the overall pulling force, with respect to the line passing through the two centers of stiffness of the LRB devices, to vanish, as follows:

$$F_{T,i} + F_{T,e} = F_T \quad F_{T,i} \times d_i = F_{T,e} \times d_e \tag{7}$$

In this way, given the values of the distances above $d_e = 5.0$ m and $d_e = 4.35$ m, the pulling force in the two cables resulted $F_{T,i} = 800$ kN and $F_{T,e} = 700$ kN.

Each steel anchorage was designed so as to remain elastic and was studied by means of a Finite Element Model employing planar triangular elements with six degrees of freedom for each node. The color maps for both the geometrical thickness of each element and the stress distribution corresponding to the application of the maximum pulling force can be found in Figure 21. The resulting steel anchorages are portrayed in Figure 22.

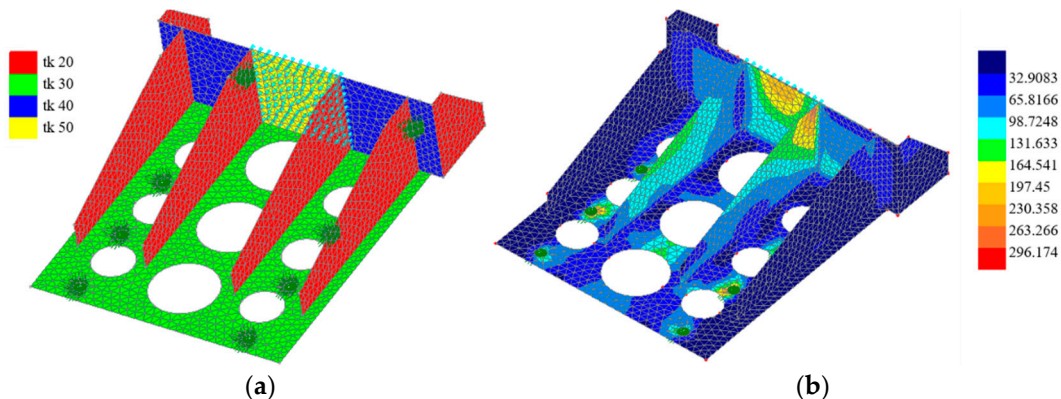

**Figure 21.** Finite Element Model (FEM) model of the anchorages: (**a**) thickness color map, and (**b**) stress field (MPa) according to the Von Mises criterion.

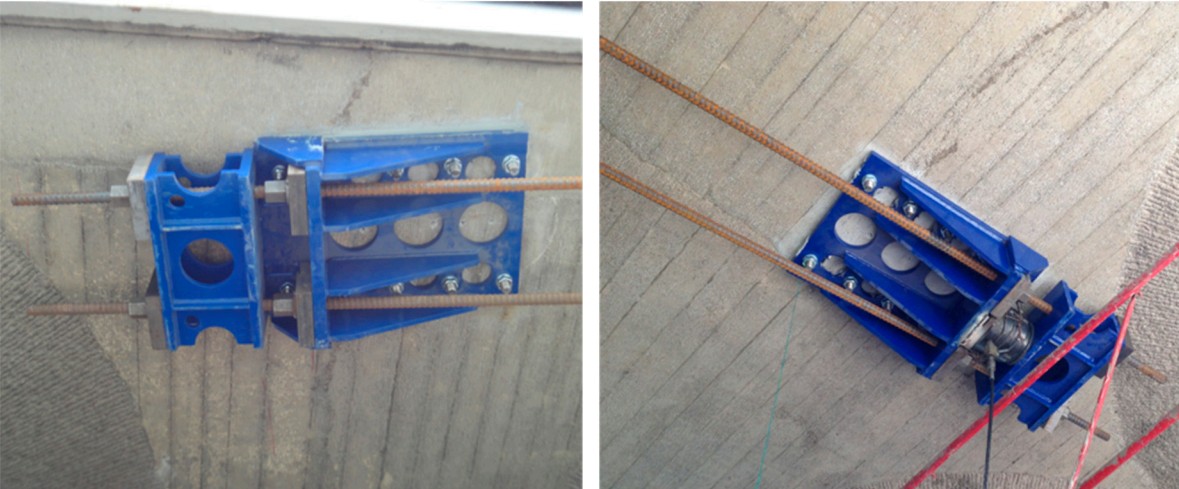

**Figure 22.** Passive (**left**) and active (**right**) anchorages.

## 4. Discussion and Further Developments

The technical solution herein presented, which was adopted to integrate base isolation and translation of the case-study strategic building, constitutes an example of advanced design that fully employed the potentialities offered by some modern technologies available in civil engineering. In fact, the adopted solution proved itself very effective in reducing the costs, fulfilling the designed constraints imposed by the Superintendence of Cultural Heritage, and maximizing the benefits in terms of highest degree of seismic rehabilitation of the existing building. For these reasons, it is a design solution that is undoubtedly worth sharing with both technical and academic colleagues. With the former ones, for its replicability, and with the latter ones, for the numerous ideas it offers in terms of both further research and refinements of the current regulatory requirements [29].

Since it is a somewhat pioneering intervention methodology which adopts and combines different and relatively innovative technologies, it presents several aspects to which due attention needs to be paid during the design. One of these aspects is represented by precisely controlling the re-centering operations by suitably foreseeing every phenomenon to be expected at the moment of removal of the temporary constraints from the LRB isolators. Among the questions to answer, in the most precise way possible in order to minimize the possibility of unexpected events occurring during the implementation, they are the ones listed hereinafter. How long can the LRBs remain deformed before the removal of the temporary constraints in order for the natural rubbers not to undergo re-crystallization?

In which way can the prolonged deformation of the lead core affect the re-centering of the isolators, taking also into account the relative cross-section dimensions with respect to the NR? What is the influence of the value of the axial load, which is carried by each LRB isolator at the removal of the constraint, on its capacity of deformation and its elastic recover? This, taking also into account either the increase or reduction of the acting axial load as a function of the moment due to the horizontal thrust applied at a certain height from the LRBs? How can the frictional sliders, in the shape of tripods in this case study, which are placed in parallel with the LRBs, affect the re-centering at the removal of the temporary constraints? For each of the tripods, which role is played by the variation of axial load as a function of the horizontal thrust exerted by the hydraulic jacks bearing on the central building?

Beyond the brilliant idea to install the pre-deformed isolators in order to also translate the lateral edifices, the re-centering operations are not straightforward. Some useful hints on some peculiar phenomena affecting the behavior of natural rubbers, which will not necessarily be comprehensive, for the sake of brevity, but that can be a useful starting point for further study on specialist textbooks. Among the phenomena that can influence the behavior of natural rubbers, there are certainly the strain-induced re-crystallization and the relaxation (e.g., [38–40]). The former was already known at the beginning of the last century (e.g., [41]), and the most important aspects were unraveled about twenty years later (e.g., [42]). However, the dependence of the crystallization on strain, temperature, and filler content have been more thoroughly investigated only in the last two decades. Researchers have also grown interested in the time-dependence of the strain-induced crystallization only recently.

Recent experimental results have shown that long-term applied deformation induces higher re-crystallization levels and subsequent grain growth, with consequent reduction of the strain hardening (e.g., [43–45]). The time-dependence of strain-induced crystallization in cross-linked natural rubber, studied using synchrotron wide angle X-ray diffraction (WAXD) [46], has shown that the lower the frequency of the dynamic loading, the higher the crystallinity becomes. Such an outcome can be yielded from the test results depicted in Figure 23a [46], where the time histories of imposed deformation $\varepsilon(t)$ and the consequent crystallinity $\phi(t)$ on a natural rubber specimen are superimposed to each other as function of time. The NR specimen had been stretched to a deformation of $\varepsilon_i(t < 0)= 365\%$, to which a value of crystallinity of $\phi_i(t < 0)= 6\%$ corresponded. Subsequently, for $0 < t \leq 5$ s, the same specimen had been subjected to a cyclic deformation, with an amplitude of $0 \leq \varepsilon(t) \leq 365\%$ and frequency of almost 1 Hz, and the consequent crystallinity had been measured stepwise. As can be seen, during the high frequency deformation, the parameter measuring the crystallinity reduced to $\phi_{max}= 100\%$. Thus, it is clearly visible that under dynamic loading, the maximum $\phi_{max}$ (the measure of crystallinity that can actually be estimated by resorting to different models, e.g., [47]) is considerably lower than the initial level $\phi_i$, independent of the same level of imposed strain. Some experiences recently conducted on full-scale devices have highlighted the relaxation effect under fixed displacement at a macro scale (e.g., [48,49]) and how the effect of prior long-term deformation may affect the characteristic strength of the LRB in seismic conditions (e.g., [50]). The graphs depicted in Figure 23b show the results of an experimental test [49] in which the isolator was subjected to a longitudinal displacement equal to 5 mm applied in 120 s (0.04 mm/s) and then kept deformed for 3600 s (1 h). At the displacement of 5 mm, the corresponding force was equal to 138 kN, and after 1 h it had reduced to 80 kN, with the decrease factor of 0.4. Looking at the trend of the function $F(t)$, we can observe an asymptotic behavior from the second hour of testing. Note that the adopted motion velocity was such as to minimize the temperature gradient that usually follows the lead deformation, thereby excluding its influence on the mitigation of the relaxation effect.

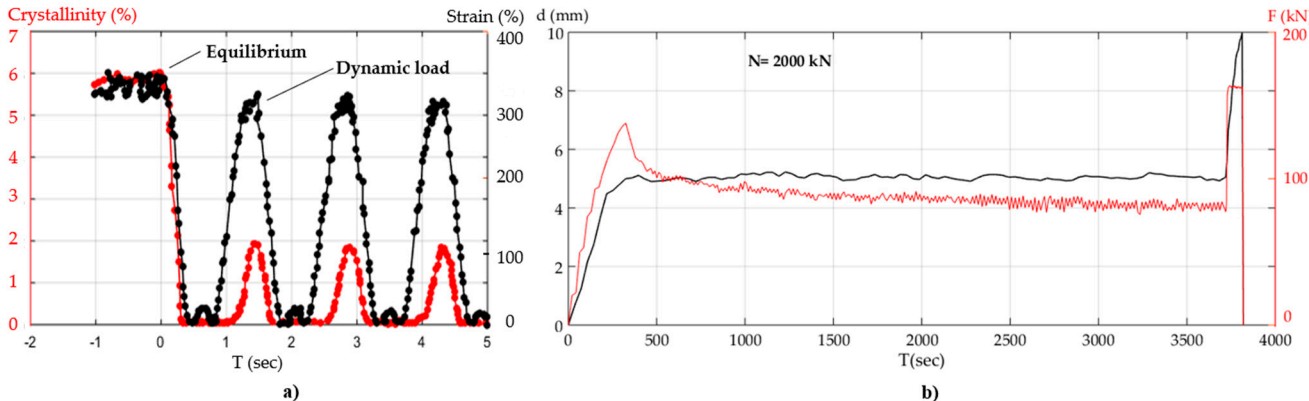

**Figure 23.** Crystallinity and Relaxation for Natural Rubbers (NRs): (**a**) crystallinity as a function of time under dynamic loading at a frequency of 1 Hz (data from [46]); (**b**) time histories of displacement (black line) and applied force (red line), for a displacement-controlled test on a LRB isolator with diameter of 45 cm (data from [49]).

In the relaxation test, whose results are shown in Figure 24a [49], a lateral displacement of 60 mm was imposed in 15 s, then the testing system was changed from force to displacement control so that the applied horizontal load dropped to zero in 5 s. After the load was removed, the displacement was monitored and recorded during the next 2 h. During such a short time scale, as expected, no NR recrystallization occurred. As shown in Figure 24a, the residual deformation after the removal of the load was roughly equal to 58 mm after 15 min and remained almost constant for the next 105 min.

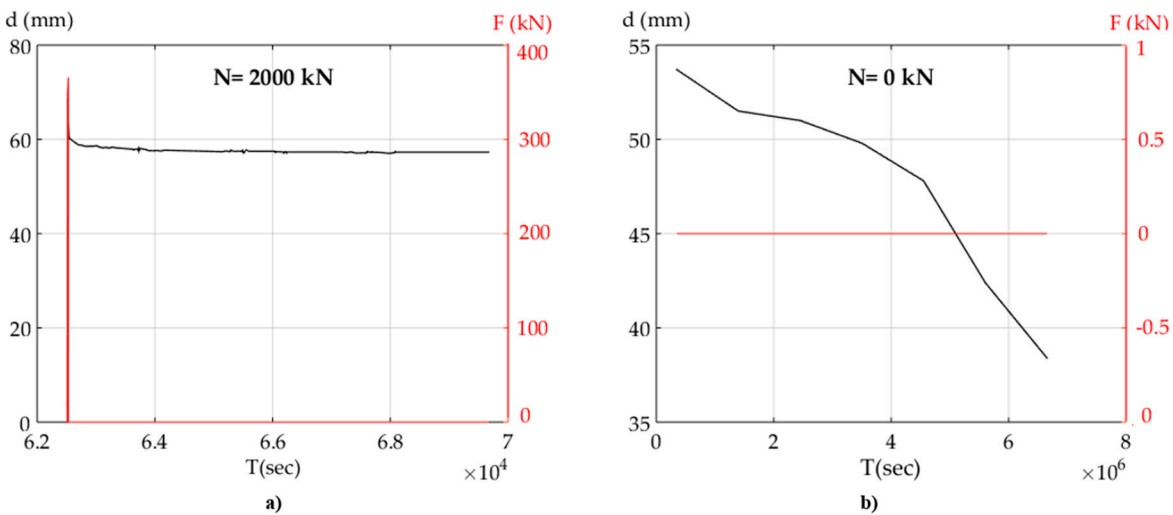

**Figure 24.** A Relaxation test [49] carried out on a real-scale LRB specimen: imposed force $F(t)$ and displacement $d(t)$ time histories for, (**a**) $62.4 \leq t \leq 69.4$ ksec and constant value of applied axial load $N_{Sd} = 2000$ kN and for, (**b**) $76.87$ ksec $\leq t \leq 6.65$ Msec and vanished axial load $N_{Sd} = 0.0$ kN (data from [49]).

As a last test, the axial design load was removed, and the displacement was monitored and recorded for the next two months. The relevant graphs are depicted in Figure 24b. As shown, the residual displacement reaches almost the expected theoretical value in about two months. Note that, as in the previous test, the horizontal force was applied in a very short time with respect to the crystallization typical time scale.

The viscous behavior of the lead core and its interaction with the NR as a function of their possible geometrical ratio also need to be taken into due consideration. Further research is desirable in this respect.

It is undoubtable that the time extents necessary to induce rubber recrystallization are much longer than those of storage of the deformed LRB isolators at the construction

yard. This is not true for the period of time the isolators remain under the columns before an earthquake occurs. In fact, the question arises if it is necessary to foresee during the maintenance phase, beyond the sterile visual inspections, the measurement of a parameter indicating the level of crystallization. Moreover, as a function of the value of crystallinity, a forced deformation should be prescribed once in a while during the building lifetime in order to guarantee that, at the occurrence of an earthquake, the LRB isolator is perfectly functioning. The sliding threshold in the sliders [51] is another aspect that researchers must be aware of, and which needs to be suitably controlled from a numerical standpoint as a function of the axial load and of the rocking effect in the columns due to the re-centering horizontal thrust.

## 5. Conclusions

The design of the strengthening interventions usually poses the difficulty of the search for a well-balanced solution between respect for the cultural heritage and guarantee of safety. The former involves the adoption of the so-called minimum intervention criterion, which often implies the acceptance of lower safety levels just to avoid too invasive interventions that may change the formal aspect of structural components. On the contrary, the latter implies the necessity to pursue the rehabilitation, mainly in such a case of a strategic building. This case study posed additional difficulties consisting of other design constraints, some of which were quite original and given by the owner of the buildings and the technicians of Florence Superintendence for Cultural Heritage, as well as from intrinsic structural problems, as better shown in the following short list:

- The work activities on the upper floors should not be interrupted during the construction site operations;
- The characteristic "mushroom" shape of the ground floor columns should not be modified, and therefore any radical strengthening interventions that would alter their shape and figurative value had to be excluded;
- The ground inter-story net height had to be kept unchanged;
- The hollow cross section of some of the columns, dedicated to housing the sewage ducts, had to be maintained;
- the technical joint between each of the lateral edifices and the central one was inadequate to accommodate the relative displacements during the out-of-phase oscillations induced by an earthquake.

For the mentioned reasons, base isolation by means of the installation of pre-deformed Lead Rubber Bearings (LRBs) was singled out as the most suitable technology, capable of fulfilling each of the design requests and providing an increasing of about 300% of the level of adequacy respect the ante-operam conditions. In fact, seismic isolation reduced the need to intervene on the existing structure, thus preserving the peculiar mushroom-shaped ground columns, which were the main concern of the Superintendence for Cultural Heritage. It allowed for rehabilitating the building in order for it to remain fully operational in the aftermath of a seismic event. The installation of pre-deformed LRB isolators allowed to seize on their elastic recover to effectively enlarge the technical joint by horizontally translating each of the lateral buildings.

The mushroom columns are hollow and house the sewage ducts. Thus, it was necessary to design an original type of slider labelled a tripod, which comprehends three pot bearings underneath three corresponding horizontal sliding surfaces, among which the necessary space for the passage of the duct is guaranteed.

Particular attention must be paid during the design to the evaluation of the residual displacement to be expected at the removal of the temporary constraints from the pre-deformed LRB isolators. In fact, from the statements in paragraph 5, there is a strong interaction between (a) the mechanical properties of the natural rubber of which the LRBs are made; (b) the mechanical properties of the lead core; (c) the mechanical properties of the frictional sliders in the tripods; and (d) the variation of the axial load in the columns, due

to the applied horizontal thrust. These interactions, which are highly non-linear, require further investigation in order to further refine this method of intervention.

However, from theoretical and experimental investigation emerged the importance of the relaxation of lead and rubber for possible future in-depth studies on relevant factors that may describe the non-linearity mentioned above, especially regarding the evaluation of the residual displacement of the pre-deformed LRB isolators subjected to temporary constraints.

The ratio between the areas of lead and rubber would seem to substantially affect the behavior of the lead rubber bearing. In fact, when this ratio is less than one-third and the strain is high (close to 100% or more), the main contribution given from rubber that is affected is high relaxation. In particular conditions, as in the case of the device tested at the University of Basilicata, significant residual displacement was recorded. This phenomenon is not clear from a mechanical point of view, but the fact that the residual displacement has reduced to the expected theoretical value when the load was been removed suggests that it may be an effect induced by the high axial load that could be defined as a "stick effect." In the light of the test results, the contribution can be so significant that it radically affects the re-centering of the device, leaving a high residual displacement.

For the relaxation factor, this phenomenon would seem to concern both materials, but only in the case of the yielding strength force are they comparable. When they are not, which happens in a majority of cases, the relaxation factor is essentially related to lead and can be evaluated as equal to 0.5, already next to one hour from the application of loads. Its value depends on the frequency and the strain rate of the rubber. As a general rule, the lower the frequency, the higher the relaxation factor. In addition, rubber gives a contribution not negligible when is strained at least up to 100%.

Possible future developments may explore experimental tests on the device with a special focus on (a) the lead/rubber areas ratio, and (b) the so-called "stick effect".

**Author Contributions:** Conceptualization, design, formal analysis, writing-original draft preparation M.V.; seismic engineering supervision G.M.; writing-review and editing, V.B. All authors have read and agreed to the published version of the manuscript.

**Funding:** This research received no external funding.

**Data Availability Statement:** The data presented in this study are available on request from the corresponding author.

**Acknowledgments:** The authors wish to acknowledge Eng. Luigi Massa, Research & Development of Tensa s.r.l., for the interesting discussions about the long-term effects concerning the LRB devices.

**Conflicts of Interest:** The authors declare no conflict of interest.

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
