# Peer review of "Integrated Solution-Base Isolation and Repositioning-for the Seismic Rehabilitation of a Preserved Strategic Building"

_buildings, doi:10.3390/buildings11040164_

Round 1

Reviewer 1 Report

The manuscript is well-written and it includes both the research findings as well as details and concerns (or limitations) in the practical applications. The case study is very useful to guide the practical engineers in seismic retrofit of structures as well as for future research study on the base isolation design. The review is really appreciating the discussion parts in various options and corresponding analyses, which bridges research with practical applications, even though the length of the whole manuscript is pretty long. Therefore, it can be accepted and published in its present format.  

Author Response

Thank you for your review report. 

Kind regards.

Reviewer 2 Report

This manuscript presents the details of the design and application of the base-isolation technique for the seismic rehabilitation of a vulnerable building. The presented study does not add much theoretical knowledge to structural/earthquake engineering. However, it deals with a real-world structure and addresses challenges associated with the application of base isolation in an existing structure. Therefore, the study is certainly valuable to the structural/earthquake engineering society. Furthermore, the case-study building is a special and essential facility in Italy that was identified as vulnerable. This further highlights the significance of the performed study. The manuscript first illustrates the weakness of the building and the need for seismic rehabilitation. Then, it provides the details of a designed base isolation system and its application. The presented approach for the installation of base-isolation in an in-service structure without interruption of the operation of the structure is the strong point of this manuscript.

Before, making a final decision regarding the manuscript, this reviewer would like for the authors to address the comments and concerns that arose when reviewing their manuscript. The detailed comments can be found on the annotated PDF file of the manuscript that is attached to this letter. These comments can be categorized into the following groups:

  • Literature review: The need to conduct a deeper literature review. For example, in lines 45-47 the authors have stated that “Smaller inertia forces, smaller inter-storey drifts, and smaller seismic forces in structural elements there-46 fore characterize base-isolated buildings.” This statement is not always true. For near-field earthquakes, or earthquakes recorded on soft soils, seismic isolation may amplify the responses instead of reducing them. It highly depends on the characteristics of the building and ground motion. Furthermore, this article deserves a much deeper literature review. This reviewer has suggested that the authors modify lines 45-47 and add a few paragraphs to clearly explain the effectiveness of base isolation for buildings of different heights subject to ground motions of different characteristics (please see the annotated PDF).

  •  Language issue and manuscript organization: In many cases, the reviewer found the manuscript very difficult to follow. Informal language, complicated sentences, grammar issues, long sentences, unclear statements are very common issues in this manuscript (even in the title of the manuscript). Therefore, the manuscript needs a much better organization and also English editing by a professional English editor. The reviewer has highlighted some (not all) of these issues in the manuscript.

  • Technical issues: In a few cases, the reviewer has concerns regarding the calculations performed in the manuscript. For example, in Section 3.2, the authors have estimated the shear force associated with the development of the plastic hinge in the columns. How do the authors know that the non-isolated building goes nonlinear? The reviewer is asking this because it seems that the design spectral acceleration for the building is quite low. Proving the inelastic behavior of a building under a design earthquake needs more advanced analysis such as push-over or response history analysis. At a minimum, the authors should roughly show that the design shear force exceeds the building yield shear force. The authors can approximately investigate this issue following these steps (a) show the 5%-damped design acceleration spectrum for the building site and the value of the design spectral acceleration at the period of the non-isolated building (meaning that the authors also need to provide the period of the non-isolated building; a rough estimation of the period based on the approximate equations available in seismic design codes is also acceptable). (b) Compute the elastic base shear = building seismic weight * 5%-damped spectral acceleration value at the building period. (c) Divide this elastic design base shear by a response modification factor (named R in ASCE/SEI 7 load standard) that could be 5-7 for a concrete building (the authors need to check the Italian code). (d) Compare this value with the yield shear force in the columns. If this value is greater than the yield shear, the building might (not certainly) experience inelastic actions during a design earthquake. Please clarify this issue or propose an alternative approach showing that the building might experience inelastic actions under a design earthquake. Another question: The shear force associated with the plastic moment in the peculiar mushroom-shaped column has been computed based on a cantilever column assumption. However, it seems that the connection of the column to the floor beams (slabs) is moment-resistant. In other words, the shear force could be 2*Mp/H instead of Mp/H assumed by the authors. Please clarify.

  • The seismic load used to evaluate the structure: The period of the isolated structure is 2.0 s. For this range of period and medium to high-intensity ground motions, the spectral displacement of the isolated structure should be quite big (e.g., 0.4 to 0.6 m). However, the authors have reported a spectral displacement of only 0.1 m. A further evaluation of the seismic force they have used in their analysis illustrates that the ground spectral acceleration response for T = 2 sec and equivalent damping = 16% is only 0.06 g (line 462 of the manuscript) which is relatively low. The authors need to show the baseline 5%-damped (design) acceleration spectrum (this was also requested in comment No. 3) and also the 16% damped spectrum. This allows seeing how base isolation has reduced the spectral acceleration value.

For detailed comments please see the annotated PDF file of the manuscript attached to this letter.

Author Response

Your suggestions about the literature review are inserted in the introduction and references.

Thank you for your review report.

Kind regards.

Reviewer 3 Report

The paper study the design of the interventions for the full seismic rehabilitation of a case-study reinforced concrete buildings located in Florence, Italy called Ante-operam. The authors tried to keep a balance between design of the interventions and the need to safeguard as much as possible the peculiar architectural elements. In order to facilitate the different challenges, some even in contrast with each other they applied the so-called minimum intervention. Another problem was the existing thermal joint between the adjacent edifices which were inadequate to prevent from pounding each other, during an earthquake. The design strategy followed by authors which fulfill all these design constraints, was based on the employment of base-isolation. The paper presents all the details of the design procedure, along with the innovative aspects and the designed devices. The proposed strategy it is promising for the application of intervention in historical buildings. Further ideas for new research and developments are also mentioned in the paper.

The paper addresses a topic posing numerical challenges and having practical significance. It is methodologically correct. The paper is suitable for publication.

The literature review in introduction is thorough and it is very well written, however some additional references listing below regarding the use of cables for rehabilitation and intervention in historical buildings and the use of structural control as another direction to protect building against earthquake excitation could be added in the introduction.

  1. Papavasileiou, G.S. & Pnevmatikos, N.G. (2017). Optimized design of steel buildings against earthquake and progressive collapse using cables. International Journal of Progressive Sciences and Technologies, 6(1), 213-220.
  2. Nikos G. Pnevmatikos, George C. Thomos “Stochastic structural control under earthquake excitations” 2014, Journal of Structural Control and Health Monitoring, Vol.21, No 4, 620-633.

Author Response

(The authors gave the same response as above.)

Reviewer 4 Report

This paper carries out a case study that investigates the integrated solution for the full seismic rehabilitation of a preserved strategic building using base isolation technologies. This study presents a unique solution to figure out several obstacles that may hardly avoid using traditional construction methods applicable to the retrofit of the building of interest. The subject dealt with in this paper meets with one of the Journal of Buildings. The contents are well-organized with rational basis and are understandable for potential reads of the journal. However, the authors should address the following comments or, if necessary, revise the manuscript to improve the quality of the submitted manuscript.

  1. Please provide the seismic capacity of the building before rehabilitation and compare it with that of the building after rehabilitation.
  2. The authors should analytically present why the target gap of 100mm to avoid the pounding effects under the design earthquake was selected.
  3. Please provide the results of prototype tests and production tests of both types of base isolators. Also present the variation in the characteristics of the base isolators obtained from the tests.
  4. Please explain the fire resistance of base isolators, if required.
  5. Please discuss the implementation of the Tripods instead of a single pendulum system.

Author Response

Thank you for your suggestions. All notes and corresponding answers have been collected in a PDF file attached to the present coverletter.

Kind regards.

Round 2

Reviewer 2 Report

This reviewer believes that the authors have adequately addressed the comments/questions which arose in the first round of review of this manuscript. However, I strongly recommend that the authors carefully proof-read their manuscript to remove the minor language issues. For example, lines 57-58 read "... for seismic protection of, but design ....". It should be "... for seismic protection, but design ....".